# Heimdallarchaea encodes profilin with eukaryotic-like actin regulation and polyproline binding

Sabeen Survery [1]✉, Fredrik Hurtig[1], Syed Razaul Haq [1], Jens Eriksson [2], Lionel Guy [2], K. Johan Rosengren[3], Ann-Christin Lindås[1] & Celestine N. Chi [2]✉

It is now widely accepted that the first eukaryotic cell emerged from a merger of an archaeal host cell and an alphaproteobacterium. However, the exact sequence of events and the nature of the cellular biology of both partner cells is still contentious. Recently the structures of profilins from some members of the newly discovered Asgard superphylum were determined. In addition, it was found that these profilins inhibit eukaryotic rabbit actin polymerization and that this reaction is regulated by phospholipids. However, the interaction with polyproline repeats which are known to be crucial for the regulation of profilin:actin polymerization was found to be absent for these profilins and was thus suggested to have evolved later in the eukaryotic lineage. Here, we show that Heimdallarchaeota LC3, a candidate phylum within the Asgard superphylum, encodes a putative profilin (heimProfilin) that interacts with $PIP_2$ and its binding is regulated by polyproline motifs, suggesting an origin predating the rise of the eukaryotes. More precisely, we determined the 3D-structure of Heimdallarchaeota LC3 profilin and show that this profilin is able to: i) inhibit eukaryotic actin polymerization in vitro; ii) bind to phospholipids; iii) bind to polyproline repeats from enabled/vasodilator-stimulated phosphoprotein; iv) inhibit actin from Heimdallarchaeota from polymerizing into filaments. Our results therefore provide hints of the existence of a complex cytoskeleton already in last eukaryotic common ancestor.

[1] Department of Molecular Bioscience, The Wenner-Gren Institute, Stockholm University, Stockholm, Sweden. [2] Department of Medical Biochemistry and Microbiology, Uppsala University, Uppsala, Sweden. [3] School of Biomedical Sciences, The University of Queensland, Brisbane, QLD, Australia. ✉email: sabeen.survery@su.se; chi.celestine@imbim.uu.se

The evolutionary events which led to the first eukaryotic cell are still controversial[1–4]. The Asgard genomes encode a variety of eukaryotic signature proteins previously unseen in prokaryotes. Functional and structural characterization of these proteins is beginning to shed light on the complexity and pedigree of the ancestral eukaryotic cell[5,6]. In eukaryotes, the key cytoskeletal protein actin is important for diverse cellular processes such as membrane remodeling and cell motility[7]. Dynamic polymerization of actin both provides structure and generates the force which drives motility and membrane remodeling. These processes demand rapid filament assembly and disassembly on microsecond timescales. In eukaryotes, a variety of highly adapted proteins including gelsolin, profilin, VASP, ARP2/3 and signaling molecules (Phosphatidylinositol-4,5-bisphosphate ($PIP_2$)) are crucial for organizing cellular cytoskeleton dynamics. Amongst others, the Asgard genomes encode predicted putative profilin homologues that regulate eukaryotic actin polymerization in vitro[5,8]. Interestingly, Asgard profilins appear to be regulated by $PIP_2$, but not by polyproline-rich motifs which are important for the recruitment of actin: profilin complexes in eukaryotes[5,9]. These findings indicate that the Asgard archaea may have possessed analogous membrane organization to present-day eukaryotes, but that polyproline-mediated profilin regulation may have emerged later in the eukaryotic lineage[5]. Here, we show that Heimdallarchaeota LC3, a candidate phylum within the Asgard superphylum, encodes putative profilin (heimProfilin) that interacts with $PIP_2$ and polyproline motifs, implicative of an origin predating the rise of the eukaryotes. Additionally, we provide evidence for a novel binding mechanism to the best of our knowledge whereby an extended N-terminal region abolishes $PIP_2$ and modulates polyproline interactions. Lastly, we provide the first evidence for actin polymerization of an Asgard actin homologue. In context, though this was an in vitro study, the findings indicate the existence of a complex cytoskeleton already in the last eukaryotic common ancestor (LECA).

## Results

### Heimdallarchaeota LC3 encodes a profilin (heimProfilin) with a probable extended N-terminus.

The recent discovery of the Asgard superphylum represents a major breakthrough in the study of eukaryogenesis[1,8]. While the cells belonging to the Asgard phyla are predicted to encode a large number of eukaryotic signature proteins (ESPs)[1,8], our knowledge of the structure and functions of these protein homologues is limited[5,10]. To verify that Heimdallarchaeota LC3 encodes a *bona fide* profilin, we determined the 3D protein structure using nuclear magnetic resonance (NMR) spectroscopy. Several profilin structures from the Asgard superphylum including Loki profilin-1, Loki profilin-2, and Odin profilin have been determined previously by X-ray crystallography both individually and bound to rabbit actin[5]. However, there are considerable phylogenetic differences separating the known Asgard phyla, and Heimdallarchaeota are currently thought to encode the most eukaryotic signature proteins (31 for Heimdall, versus 25, 25, and 28 for Odin, Thor, and Loki, respectively of the 38 that Eukaryotes express) than any other Asgard phyla[8,11]. Nevertheless, sequence conservation amongst the Asgard profilin homologues is relatively low and identity is mostly established through structural homology. At first glance, our NMR structure depicts a typical profilin fold, with seven strands interlinked by loops connecting four helices (Fig. 1 and Supplementary Fig. 1). However, the orientation, positions, and length of the helices and loops differ dramatically compared with Loki profilin-1 and canonical eukaryotic profilins. It should be mentioned that the structures of the Lokiarchaeota profilins were described recently

and it was found that the main structural difference between the Loki and eukaryotic profilins was the presence of an extended loop called the Loki-loop[5]. Our detailed structural comparison reveals that Heimdallarchaeota LC3 profilin (heimProfilin) is divergent from the Loki profilin-1[5] (root mean squared deviation (RMSD) > 3.75 Å). Notably, differences include the formation of an additional helix between residues H123–S129, the re-orientation of the N-terminal helix (residues S27–Q35) to an open conformation, a shorter Loki-loop, the absence of a helix between residues G72–P75, and the presence of a long disordered N-terminal extension (residues 1–20) (Fig. 1). It should be noted that, although the gene prediction algorithm used for the Heimdallarchaeota LC3 metagenome (prodigal) is usually very accurate to predict correct start codons, we cannot exclude that this protein is actually translated from another start site, e.g. the ATG codon located at residue 24[12]. It is also possible that both start codons are used, with various frequencies. However, in summary, structural differences indicate that despite the overall profilin fold, heimProfilin differs from the eukaryotic and the recently determined Loki profilins.

### HeimProfilin inhibits rabbit actin polymerization in vitro.

The extended N-terminal region in heimProfilin is not expressed in all Loki profilins or eukaryotic profilin but was found to be more dynamic relative to the rest of the protein chain based on the heteronuclear Overhauser effect[13]. To investigate the function of this region, we cloned and expressed a truncated form of heimProfilin which we called ΔN-heimProfilin which lacked the extended N-terminal region (residues 1–23). The overall fold of ΔN-heimProfilin was similar to that of heimProfilin as estimated from NMR backbone $^1H–^{15}N$ correlation spectra and TALOS-N helical propensity prediction (Supplementary Fig. 2 and 3). The inhibition of actin polymerization by profilins from Lokiarchaeota, Odinarchaeota, Thorarchaeota, and Heimdallarchaeota LC2 was previously described and indicated that Archaea profilins showed a measurable ability to spontaneously inhibit rabbit actin nucleation albeit not to the same extent as the human profilin[5]. To further investigate the functions of both Heimdallarchaeota LC3 wildtype and mutant profilins, we allowed rabbit actin to polymerize in the presence of heimProfilin or ΔN-heimProfilin and observed the resulting filament network with Total Internal Reflection Fluorescence (TIRF) time-resolved microscopy. In these experiments, heimProfilin (100 μM) was able to modulate the speed of filament elongation (1400 ± 480 compared to its absence 1900 ± 1150 monomers per second respectively) (Figs. 2 and 3). This result was comparative to the previous rabbit actin inhibition by Loki profilin−1 and −2[5]. In contrast, ΔN-heimProfilin (100 μM) appears to only slightly alter the filament network (Fig. 3) formation but not the elongation speed (1900 ± 1200 compared to its absence 1900 ± 1150 monomers per second, respectively) (Figs. 2 and 3). To verify these results, we followed the polymerization dynamics of pyrene labeled rabbit actin in the presence of heimProfilin or ΔN-heimProfilin. In line with the microscopy data, we found that heimProfilin was able to inhibit rabbit actin nucleation in a concentration-dependent manner (Fig. 2d–f). Conversely, ΔN-heimProfilin did not alter rabbit actin polymerization (Fig. 2e–f). The inhibitory effect of heimProfilin on rabbit actin polymerization was similar to that previously observed for lokiProfilins but slightly lower than that for humanProfilin[5].

### Heimdallarchaeota LC3 encodes actin that polymerizes into filaments.

To further verify the actin inhibitory effect of heimProfilin, we cloned, expressed, and purified an actin homologue

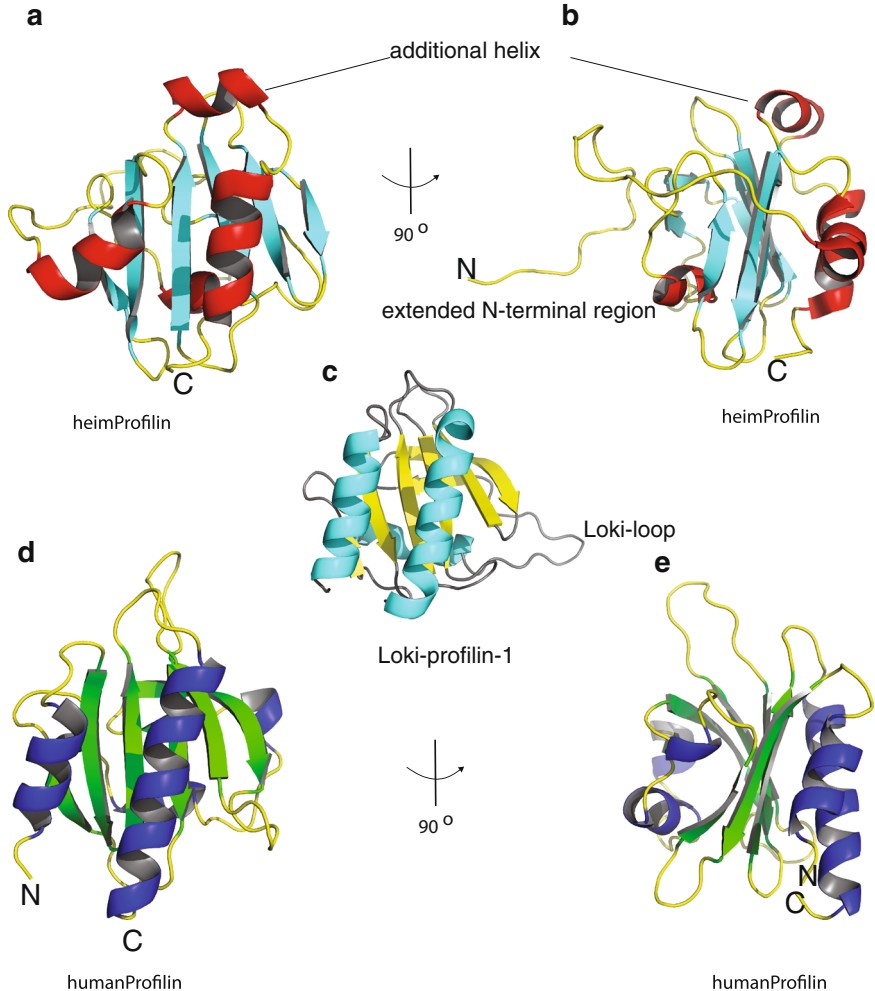

**Fig. 1 Heimdallarchaeota LC3 encodes profilin with extended structures. a** Structural representation of heimProfilin. The orientations of the N-terminal and C-terminal helices are displayed. In addition, the helix between residues H123–S129 is also shown. **b** Reorientation of the structure in (**a**) by 90° to show the extended N-terminal region between residues 1–24. **c** Structural representation of Loki profilin-1 (PDB ID:5zzb) showing the Loki-loop. **d**, **e** Structural representation of human profilin-1 (1fil) reoriented in a similar fashion for comparison to the heimProfilin in (**a**) and (**b**). The human profilin does not harbor the N-terminal extension and the C-terminal helix is slightly longer. The structural statistics are given in Table 1 and Supplementary Fig. 14. The structural coordinates have been deposited in the Protein data bank with PDB ID: 6YRR.

from Heimdallarchaeota LC3 and AB_125. Heimdallarchaeota AB_125 actin lacks the last 35 C-terminal amino acid residues, hence we named ΔC-heimActin as opposed to the Heimdallarchaeota LC3, which we called heimActin. It should be noted that the sequence identity between AB_125 and LC3 is 100%, except for the last C-terminal residue not present in AB_125. These last 35 C-terminal residues are crucial for actin polymerization[7,14]. We observed that heimActin migrated on the SDS-PAGE as a 70 kDa protein while the ΔC-heimActin migrated just around the 49 kDa marker (Supplementary Fig. 4). This was a bit surprising as the main difference between them was only 3.5 kDa. To confirm the molecular weight, we analyzed heimActin using analytical size-exclusion chromatography. We found that in the presence of a reducing agent, heimActin eluted well around a 44 kDa standard, implying that its actual size is around 44 kDa (Supplementary Fig 5). Finally, MALDI-TOF of the purified heimActin was used to confirm its identity (Supplementary Fig. 5d). Further, electron microscopy showed that purified heimActin could form thin filaments while the deletion mutant could not (Fig. 2b, c). Further, heimActin displayed a higher ATP hydrolysis activity than the ΔC-heimActin, likely a

result of losing sections of the polymerization interface (Fig. 2g). These results also provide the first evidence for polymerization of an Asgard actin. We then compared the co-sedimentation profiles of heimActin and ΔC-heimActin in presence of heimProfilin and ΔN-heimProfilin. We observed that both profilins were efficient in keeping monomeric heimActin in the supernatant (Supplementary Fig. 4a–b and j–k). The presence of the profilins along with heimActin in the pellets in the co-sedimentation imply that, besides the usual interaction of the profilins with the monomeric actin, they also interact with the heimActin filaments. In addition, we found the small amount of ΔC-heimActin present in the soluble fraction when co-sedimented together with heimProfilin (Supplementary Fig. 4c), meaning that the C-terminal 35 amino acid residues in heimActin are not the only requirement for the interaction between heimProfilin and heimActin. In comparison, co-sedimentation of heimProfilin or ΔN-heimProfilin with rabbit actin, results only in a small amount of co-sediment, implying that heimProfilin interacts mainly with the monomeric rabbit actin (Supplementary Fig. 4d–e). These results corroborate the above nucleation inhibition of rabbit actin by heimProfilin but also shows that heimProfilin interacts with heimActin filaments.

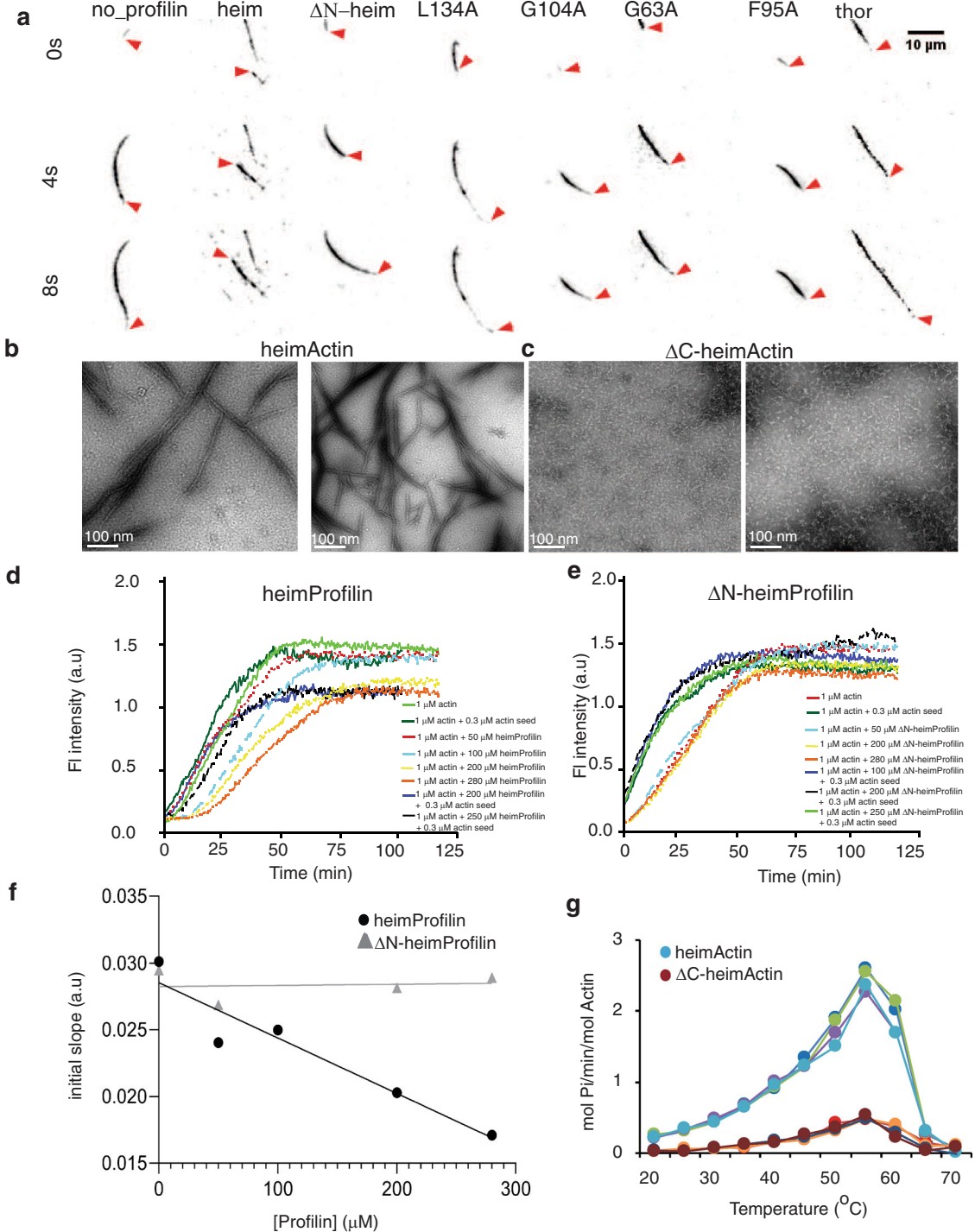

**Fig. 2 Heimdallarchaeota encoded actin (heimActin) and rabbit actin polymerization are modulated by heimProfilin.** Time dependent Total Internal Reflection Fluorescence (TIRF) Microscopy of (**a**) 1 μM of rabbit actin labeled with 0.3% Alexa Fluor 568 and 0.2% Biotin-actin supplemented with 100 μM of heimProfilin, profilin mutants or thorProfilin. **b**–**c** Electron microscopy (EM) images (at 100 nm scale) of heimActin (**b**) forming thin, uniform filamentous polymers and ΔC-heimActin (**c**) forming irregular, amorphous structures. Scale bar: 100 nm. **d** Pyrene-polymerization profiles of 1 μM rabbit actin (2% pyrene-labeled) alone or supplemented with 0.3 μM actin seeds with different concentrations of heimProfilin; 1 μM actin alone (light green), 1 μM actin with 0.3 μM actin seed (dark green), 50 μM (red), 100 μM (cyan), 200 μM (yellow), 280 μM (orange), 200 μM (blue) with 0.3 μM actin seed, and 250 μM (black) with 0.3 μM actin seed. **e** Pyrene-polymerization profiles of 1 μM of rabbit actin (2% pyrene-labeled) alone or supplemented with 0.3 μM actin seeds with different concentrations of ΔN-heimProfilin; 1 μM actin alone (red), 1 μM actin with 0.3 μM actin seed (dark green), 50 μM (cyan), 200 μM (yellow), 280 μM (orange), 100 μM (blue) with 0.3 μM actin seed, 200 μM (black) with 0.3 μM actin seed and 250 μM (light green) with 0.3 μM actin seed. **f** initial slopes of the pyrene polymerization assays in (**d**) and (**e**) plotted as a function of heimProfilin and ΔN-heimProfilin concentrations. The slopes were obtained from the linear phase of the reaction after removing the lag phase. **g** ATP hydrolysis during polymerization of heimActin (cyan, green, purple and blue) or ΔC-heimActin (brown, red, orange, and dark-red) as a function of temperature. Four replicates from each experiment is plotted as function of temperature.

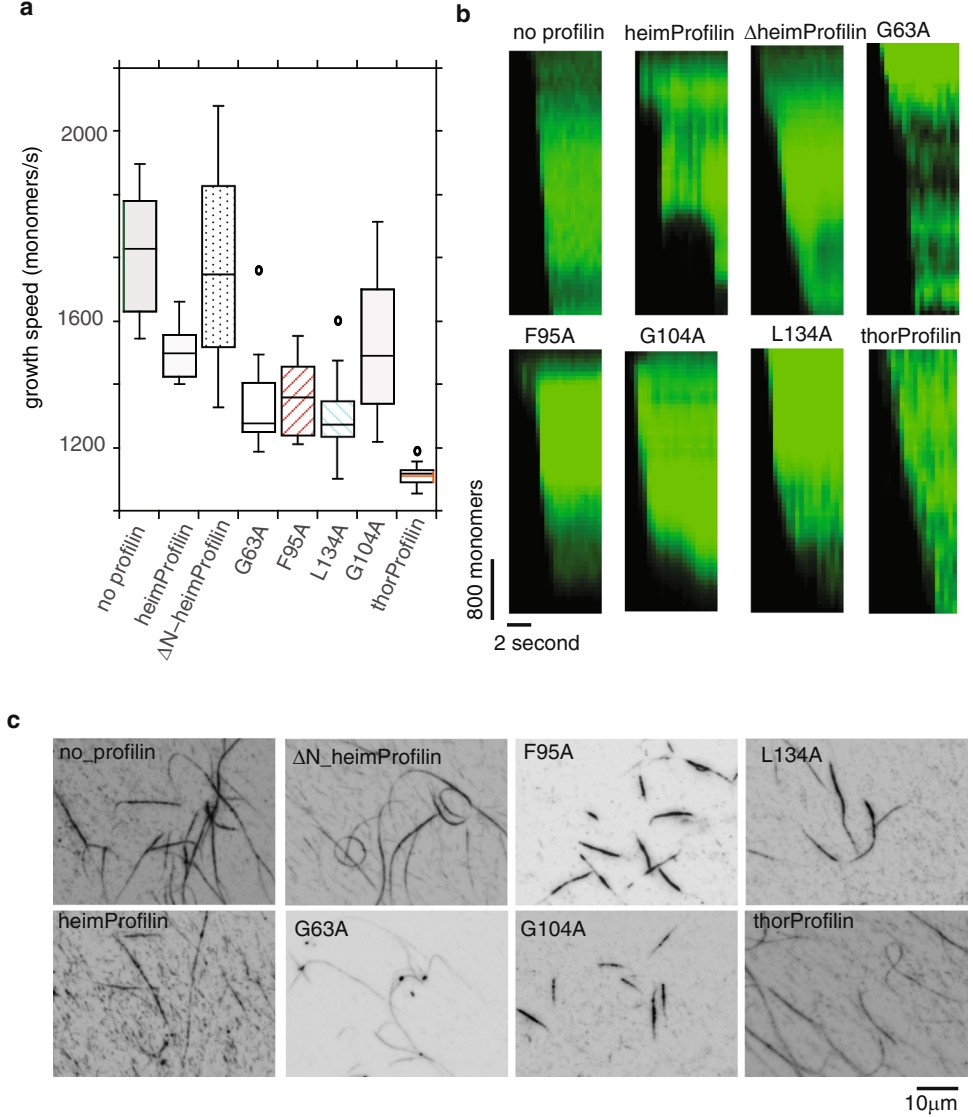

**Fig. 3 Heimdallarchaeota profilin and mutants modulate the speed of rabbit actin elongation. a** growth velocity expressed as monomers per second. Growth velocity was determined from the growth of 6–13 visible time lapse filaments. The variation of the speeds is represented with the box plots. **b** kymographs of filament elongation. **c** TIRF microscopy images of elongated filaments in the absence or presence of 100 μM heimProfilin, mutants, and thorProfilin. A detailed description of how the data was analyzed is described in the methods section. The spread in (**a**) is determined from the range of 6–13 different speeds.

This finding indicates that Heimdallarchaeota LC3 possess profilins that are able to regulate both heimActin and eukaryotic actin polymerization.

**Heimdallarchaeota LC3 encodes actin that interacts with heimProfilin.** To further investigate the interaction between heimProfilin and heimActin and to see which residues are required for the interaction, we turned to NMR spectroscopy and performed binding titrations between heimProfilin, heimActin, and their respective mutants. We observed chemical shift changes for the interaction between both heimProfilin and heimActin, in line with the sedimentation assays (Supplementary Figs. 6 and 7). We also observed that ΔN-heimProfilin interacted with heimActin in agreement with the co-sedimentation assay. HeimProfilin and ΔN-heimProfilin seem to exhibit similar chemical shift changes in the presence of heimActin when comparing identical amino acid residues at the same concentrations (Supplementary Fig. 6a). In contrast, the interaction between

ΔN-heimProfilin and ΔC-heimActin is not as extensive as for the interaction between heimProfilin/heimActin or ΔN-heimProfilin/heimActin (Supplementary Fig. 8) (fewer residues exhibiting small chemical shift changes at similar concentration). From the NMR titrations, we were able to map the site of interaction (Supplementary Figs. 6 and 7), which corresponded to the following residues in heimProfilin: N31, N36, Y52, S64 V66, R68, Q69, M71, V79, F84, R88, F95, V96, G103, G104, I106, N113, F117, T127, G130, L134, H137. For comparison, the residues responsible for actin interaction in eukaryotes are D87, R89, K91 V119-G121, G122, N125, K126, and Y129. A clear depiction of this in the form of structural alignment of the corresponding structural elements is provided in Supplementary Fig. 6. In comparison to the eukaryotic profilin-actin binding residues, we observed that these residues seem to fall into four main clusters. To verify the influence of this position on actin polymerization, we selected four residues corresponding to G63, F95, G104, and L134 from these clusters and made single-point alanine mutations at these positions. First, we verify by circular dichroism that all

mutants were well folded and stable. We then monitored rabbit actin polymerization in the presence of these mutants. We observed that G63, F95, and G104 were important for inhibition of filament elongation as mutation of these residues to alanine partly restore the speed of actin polymerization similar to the situation when no heimProfilin was present ($1660 \pm 870$, $1620 \pm 740$, and $1570 \pm 440$ vs. $1900 \pm 1150$, monomers per second, respectively). L134A ($1480 \pm 470$ monomers per second) displayed activity similar to heimProfilin. In addition, we observed that the filaments in the presence of F95A and G104A were considerable shorter (Fig. 3 and Supplementary Movie 1). Together these results indicate that Heimdallarchaeota profilin is functional and is consistent with dynamic barbed end binding of actin and that F95 in heimProfilin is important for actin polymerization.

**HeimProfilin interacts with polyproline motifs from Ena/VASP.** Polyproline motifs from the enabled/vasodilator-stimulated phosphoprotein (Ena/VASP) family of proteins are important for nucleation and elongation of actin filaments[15]. Polyproline motifs are also widespread in Asgard archaea: for example, a protein (Genbank: *OLS24758*) found in Heimdallarchaeota LC3 encodes a PPAPRPLP motif; another protein (Genbank: *TFG13563*) from Thorarchaeota isolate Tekir-14 – which contains the thorProfilin analyzed here encodes a PPPAPP motif. Other proteins in other genomes contain over 20 polyproline motifs. To verify if heimProfilin binds to polyproline we performed binding experiments both by NMR spectroscopy and isothermal titration calorimetry (ITC), using heimProfilin and ΔN-heimProfilin and polyproline motif from VASP previously shown to interact with eukaryotic profilins[9]. We observed a moderate binding of ΔN-heimProfilin to polyproline with an affinity constant of $0.30 \pm 0.10$ mM, and a very weak binding for heimProfilin with an affinity constant of $3.3 \pm 1.0$ mM (Fig. 4). The $K_D$ determined for the interaction between human profilin-polyproline interaction has been approximated to range between $0.09–0.25$ mM indicating that ΔN-heimProfilin interacts with polyprolines with similar affinities[15]. In addition, NMR monitored titration reveals that the residues responsible for polyproline binding were K22, G49, Y52, W53, I106, A111, A145, F147, and Q148 (Supplementary Fig. 9). Two residues, W53 (W32 in human) and F147 (H134 in human) correlate to that in the eukaryotic profilin, indicating that the binding interface between the two might overlap. Revisiting the structure of heimProfilin and comparing it with that of eukaryotic profilin, reveals that the N-terminal helix is orientated upwards creating a pocket that allows the polyproline motif to bind in a fashion slightly different from human profilin-polyproline binding (Supplementary Fig. 9). These striking observations explain the reason why Loki profilin-1, -2, and Odin profilins could not interact with polyproline motifs, whereas heimProfilin could. Structural data from Loki profilin-1, -2, and Odin profilins indicated that their N-terminal and C-terminal helices are parallel and closer to each other making this type of interaction highly unlikely[5]. These results suggest that, contrarily to what was previously thought, polyproline binding (to profilin) could have emerged before the split between the Asgard and Eukaryotes. This poses an interesting question, why does profilin from Loki- and Odinarchaeota not possess the N-terminal loop extension? Possibly some Asgards had this loop but lost it, or conversely, that Heimdallarchaea LC3 acquired the loop independently by convergent evolution, or through horizontal gene transfer.

**A small fraction of Asgard archaea encode profilins with N-terminal extension.** We performed a PSI-BLAST search for profilin homologs and retrieved 256 sequences from Asgard archaea, but also, more surprisingly, 8 sequences from Bathyarchaeota and one from Euryarchaeota. Although it is difficult to exclude that these proteins truly come from non-Asgard archaea, their distribution on the tree suggests that these 9 sequences are either incorrectly attributed to these organisms, contaminations, or the result of horizontal gene transfers. A maximum-likelihood phylogeny reveals a complex evolutionary history, including multiple events of duplications and deletions (Supplementary Fig. 10). Consistent with previous publications suggesting that Heimdallarchaeota might be the group most related with eukaryotes, eukaryotic profilins are nested in a clade consisting mostly of sequences from Heimdallarchaeota, although the bootstrap support for that clade were not very high (76). The group of heimProfilins most closely related to Eukaryotes consists of four sequences, one of which is the one investigated in this contribution (*OLS22855.1*). Out of the 256 profilin homologs found in the Asgard archaea, 12 had an N-terminal extension longer than 5 amino-acid residues, the longest of which was 25 residues (*MBD3350309.1*) and the second-largest was the heimProfilin investigated in this contribution (*OLS22855.1*) (Supplementary Fig. 11). Of these 12 sequences, 2 belonged to Heimdallarchaeota, 3 to Thorarchaeota, and 7 to Lokiarchaeota. Apart from a group of three lokiProfilins groupings together, they were spread out throughout the tree, strongly suggesting independent events (Supplementary Fig. 10).

**Thorarchaeota encodes profilin that interacts with polyproline.** In an attempt to see if any of these profilins retain the profilin fold and if the N-terminal extension is indeed present in the 3D structure, we first modeled the 3D structure of one member of the Thorarchaeota profilin, TFG12995.1 containing 22 amino acids upstream the known start position using the RaptorX software[16]. Indeed, we found that the overall fold of the predicted structure matches our 3D structure determined for heimProfilin with an additional N-terminal extension (Fig. 5). To further verify these results, we cloned, expressed, and performed NMR $^1$H–$^{15}$N TROSY-HSQC binding experiments of polyproline with thorProfilin and ΔN-thorProfilin (N-terminal 18 amino acid residues deleted) from TFG12995.1. We observed chemical shift changes for the interaction with polyproline for both the full-length protein and ΔN-thorProfilin with affinity measured ranging between $250–500$ μM (Fig. 6). In addition, we found that residues present at the N-terminus of thorProfilin also exhibited chemical shift changes upon binding to polyproline. Finally, we verified the effect of thorProfilin on actin polymerization by monitoring the speed of actin filament elongation using TIRF microscopy. We observed that thorProfilin (100 μM) was able to slow the speed of rabbit actin elongation substantially compared to its absence ($1105 \pm 590$ vs. $1900 \pm 1150$ monomers per second, respectively) (Fig. 3). Together, these results indicate that, these N-terminal extensions is present in other archaea and might play similar roles as observed for heimProfilin.

**HeimProfilin interacts with phosphatidylinositol-4,5-bisphosphate, with the N-terminal extension potentially regulating the interaction.** In eukaryotes membrane phospholipids, particularly phosphatidylinositol-4,5-bisphosphate (PIP$_2$), regulate the activities of many actin-binding proteins including profilin, cofilin, ezrin, Dia2, N-WASP, and meosin[17]. It should be noted that Asgard archaea probably not possess similar eukaryotic membrane architecture. However, they do express membranes with lipids that have similar features. For example, some archaeal lipids have similar inositol head groups but varied tails known as archaeols[18]. Therefore, we verified whether heimProfilin is able to

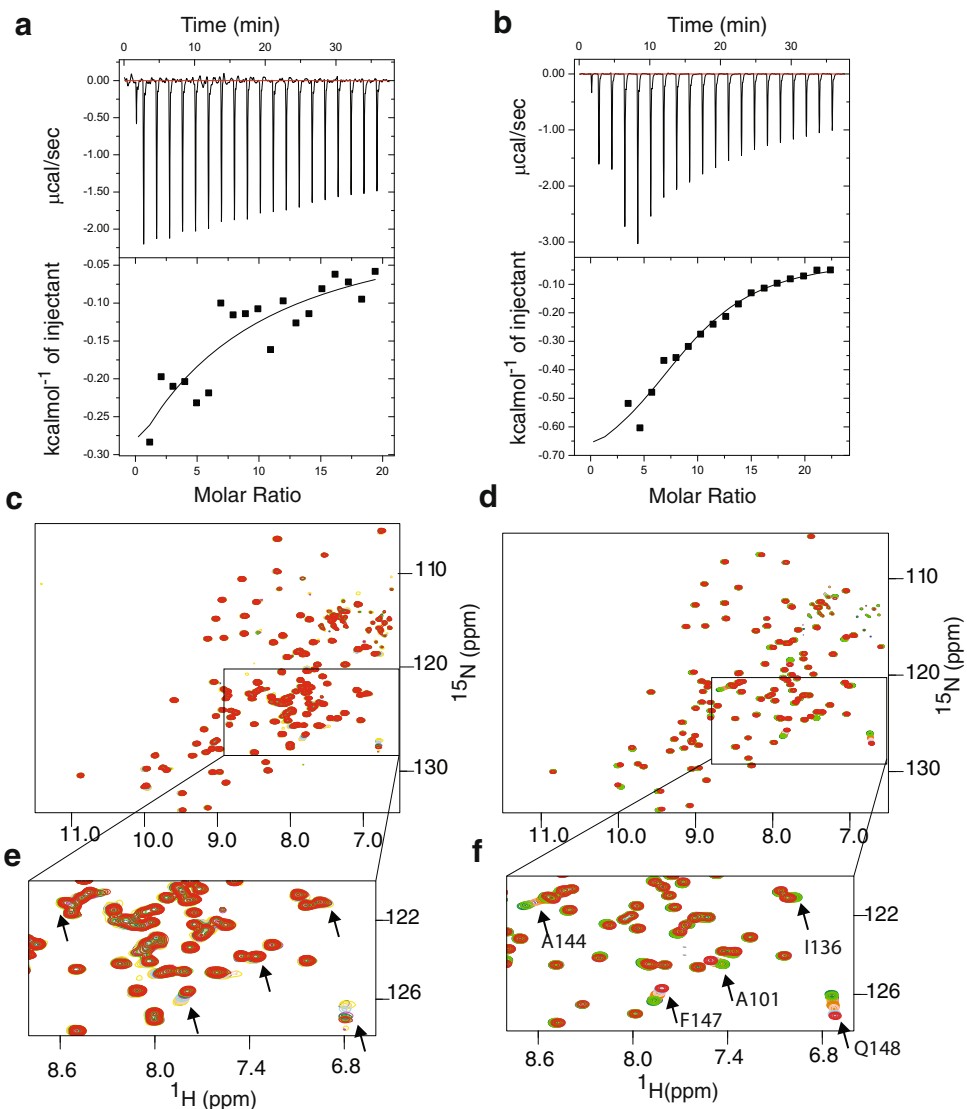

**Fig. 4 Heimdallarchaeota profilin interacts with polyproline.** Isothermal titration calorimetric (ITC) binding measurements between heimProfilin (**a**) or ΔN-heimProfilin (**b**) with polyproline motif of VASP (GAGGGPPPAPPLPAAQ). The heimProfilin showed weaker binding strength compared to ΔN-heimProfilin which had a $K_D$ of 3.3 ± 1.8 mM and 0.3 ± 0.10 mM respectively. **c, d** Nuclear magnetic resonance (NMR) $^1$H–$^{15}$N chemical shifts for the binding reaction of a fixed concentration of heimProfilin (200 μM) and ΔN-heimProfilin (400 μM) with increasing concentrations of polyproline from VASP (PPPAPPLPAAQ) respectively. For heimProfilin, the following VASP concentrations were used; 0 μM (red), 100 μM (light green), 300 μM (green), 800 μM (magenta), 1800 μM (cyan), 2300 μM (purple), 2800 μM (yellow), and 3300 μM (blue). For ΔN-heimProfilin; 0 μM (red), 100 μM (magenta), 500 μM (cyan), 1300 μM (purple), 1600 μM (orange), 2000 μM (yellow-green), 2500 μM (light red), 3000 μM (yellow), 3500 μM (blue), 4000 μM (light green) and 4300 μM green. **e, f** Expansions from (**c**) and (**d**) showing the shift of some interacting residues as the concentration of polyproline increases.

interact with PIP$_2$ by monitoring changes in NMR chemical shifts upon addition of PIP$_2$ into a solution of heimProfilin or ΔN-heimProfilin (Fig. 7). We observed that while ΔN-heimProfilin interacted with PIP$_2$, little or no interaction was observed for heimProfilin based on NMR fast chemical shift exchange (Fig. 7f–h). We estimated the affinity of this interaction (for the interaction between ΔN-heimProfilin and PIP$_2$) from the NMR chemical shift to be between 0.5–1 ± 0.3 mM (Supplementary Fig 12). Also, because the interaction between the heimProfilin and PIP$_2$ resulted in broadening of a few peaks, we reasoned that this broadening might mask the interaction and as a result are not visible by a mere chemical shift drifting as observed in the case of ΔN-heimProfilin-PIP2 interaction, (resulting in these residues to resonate in the intermediate exchange region). To probe this further, we determined $R_2$ rates for the backbone $^1$H–$^{15}$N pairs. $R_2$ rates higher than the average is often a result of $R_{ex}$ ($R$

exchange) and will result in broadening. We used a pulse program that allowed for exchange rates slower than approximately 5 milliseconds to be quenched, and those faster than 5 milliseconds which contributes to $R_{ex}$ to be observed[19]. We then compared the exchange contribution of heimProfilin and ΔN-heimProfilin free and bound to PIP$_2$. We noted that there was little exchange contribution to $R_2$ for heimProfilin, free/bound as compared to ΔN-heimProfilin free/bound, indicating that only a little or no interaction occurs between heimProfilin and PIP$_2$ (Supplementary Fig. 13) and that ΔN-heimProfilin is the main active form for the interaction with PIP$_2$. The residues responsible for phospholipid binding in the Asgard archaea have only been speculated from surface charge distribution[5]. These NMR titration experiments gave us an opportunity to map the binding interface. The following residues were observed to display chemical shift perturbation upon addition of PIP$_2$: S27, D28, L30, N31, Q35, S36,

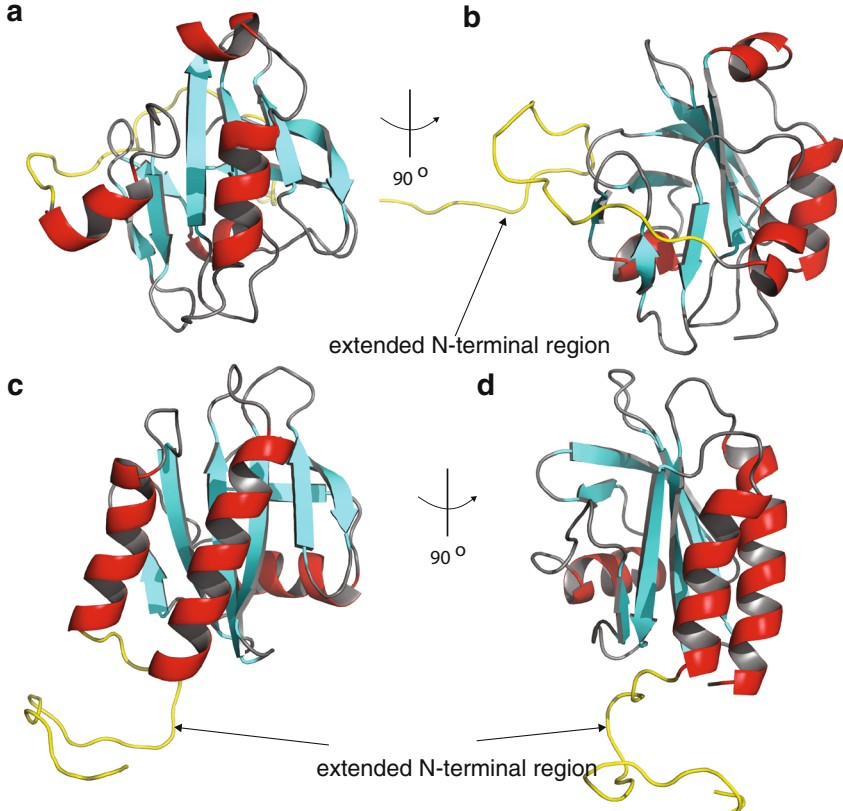

**Fig. 5 The N-terminal extension is also present in other Asgard archaea.** Structural representation of heimProfilin **a–b**, and models of Thor_TFG12995.1-thorProfilin **c–d**, showing the N-terminal extensions. The Thorarchaeota profilin was modeled in RaptorX online software while the heimProfilin 3D structure was determined by NMR spectroscopy in this study.

V43, G49, N99, K110, A111, F117, L118, S119, E139, I140, M142, M143, K146, F147, Q148 (Supplementary Fig. 12). Although the chemical shift of specific residues does not imply a direct interaction of that residue, we observed that all the residues displaying large changes in chemical shift were located on the same surface of the protein (Fig. 7 and Supplementary Fig. 12), indicating a potential binding interface for $PIP_2$. In eukaryotic profilin, K69 and K90 (K60 and K71 in Loki and K58 in Odin) were predicted to be responsible for this $PIP_2$ interaction[5]. Our NMR analysis revealed that, K110 and K146 in heimProfilin are partially responsible for this interaction (Supplementary Fig. 12). We also investigated the potential interaction of inositol trisphosphate ($IP_3$), a second messenger signaling molecule resulting from the hydrolysis of PIP2. However, we did not observe any chemical shift change upon the interaction of IP3 with either heimProfilin or ΔN-heimProfilin (Fig. 7e, f, h), indicating weak or no interaction.

## Discussion

The general consensus is that eukaryotes evolved from the fusion of an archaeal host and an alphaproteobacterium[20]. However, the exact nature of the cellular biology of either organism and/or how this event fosters the larger part of complex life still remains unclear. The recent discovery of the Asgard superphylum, the closest known prokaryotic relative of eukaryotes has changed the conversation and open up more possibilities to trace these ancestral lines[8]. The Asgards archaea (Lokiarchaeota, Thorarchaeota, Odinarchaeota, Heimdallarchaeota Wukongarchaeota, Hodarchaeota, Kariarchaeota, Hermodarchaeota, Gerdarchaeota, and Baldrarchaeota) genomes encode close actin homologues and

also several actin-binding proteins; including profilin, that regulate the actin cytoskeleton in eukaryotes. In eukaryotes, actin dynamics is also intricately regulated by several factors including phospholipids and proline-rich repeats[9,17,21]. While the actual cellular organization of the Asgard membrane is likely different from those of their eukaryotic counterpart especially in terms of the type of phospholipids expressed, their cellular membrane contains lipids with similar head groups as those found in eukaryotes[18], and they also harbor polyproline motifs.

First, we show that a specific isolate of Heimdallarchaeota (LC3), encodes a *bona fide* profilin, referred to as heimProfilin, which has many characteristics of eukaryotic profilins. We confirm that heimProfilin, as other profilins from Asgard archaea, inhibit the polymerization of rabbit actin, in vitro[10]. We also show that Heimdallarchaeota LC3 encodes functional actin, which polymerizes into filaments. Further, we show for the first time to the best of our knowledge that the heimProfilin interacts with the heimActin, demonstrating that Heimdallarchaeota LC3, as eukaryotes and likely as other Asgard archaea, regulate actin polymerization with profilin.

Second, we demonstrate that heimProfilin interacts with phospholipids, further strengthening the hypothesis that phospholipids are involved in actin modulation in Asgard archaea. Third, we show that heimProfilin and another profilin from a Thorarchaeota (thorProfilin) both interact with polyproline motifs. Polyproline motifs are present in both genomes, and are commonly found in Asgard archaea, also highlighting a potential role of these motifs in regulating actin in Asgard archaea. The structures and actin regulatory properties of profilins from other members of the Asgard phyla were determined recently[5]. It was found that, Loki profilin-1 and -2, Odin and Heimdall LC2

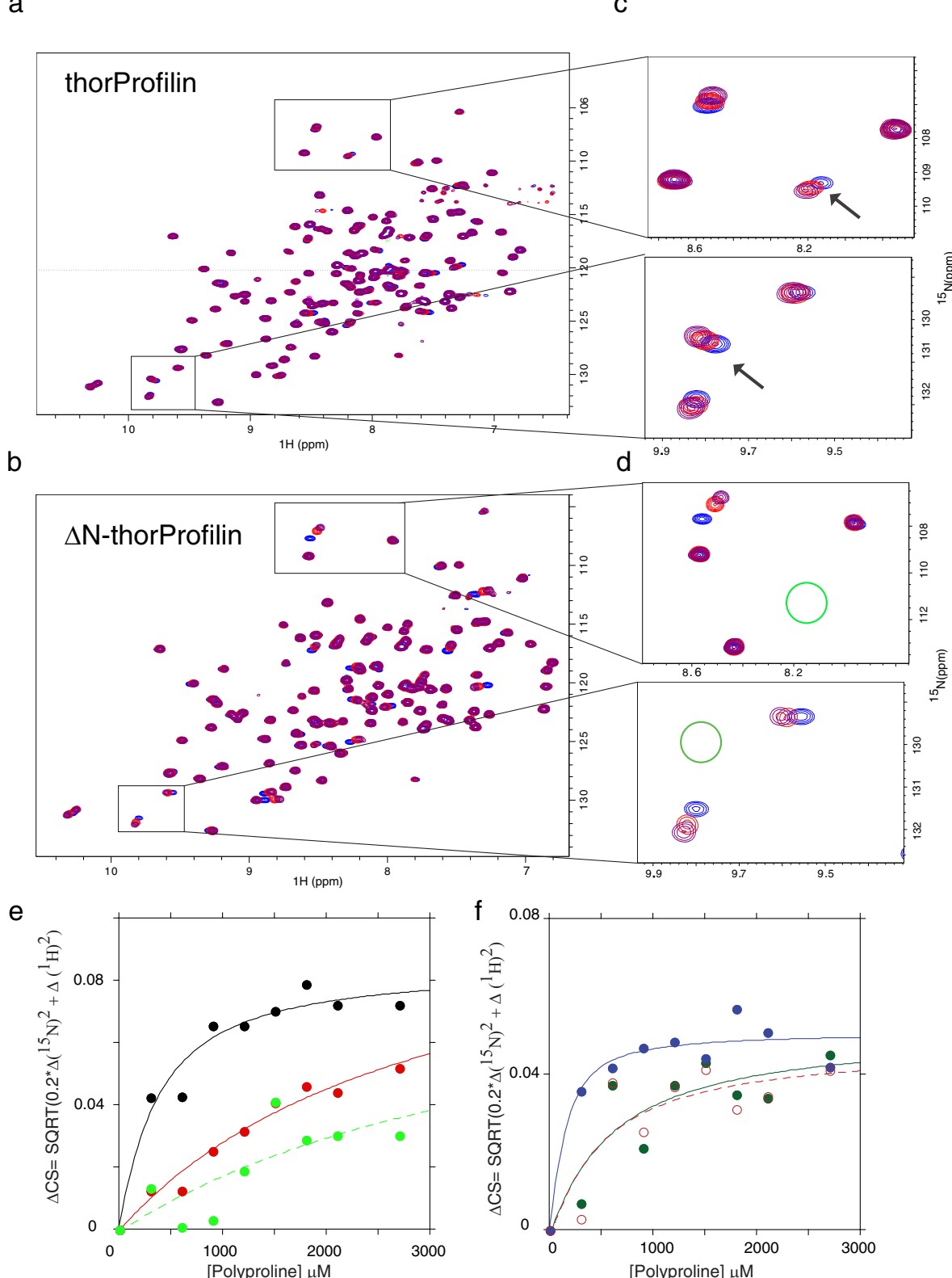

**Fig. 6 Thorarchaeota profilin interacts with polyproline.** NMR monitored $^1$H–$^{15}$N chemical shifts for the binding reaction of a fixed concentration of thorProfilin (200 μM) (**a**) and/or ΔN-thorProfilin (**b**) (200 μM) with different concentrations of polyproline from VASP (PPPAPPLPAAQ) respectively. The following VASP concentrations were used; 0 μM (blue), 5 mM (red), 7.0 mM (purple). **c**, **d** Expansions from (**a**) and (**b**) showing the shift of some interacting residues as the concentration of polyproline increases. The arrows in (**c**) indicate residues from the N-terminal extension interacting with polyproline. These residues are absent in the ΔN-thorProfilin as indicated with the green circle in (**d**). **e**, **f** Chemical shift plotted as a function of polyproline (VASP) concentration for a few residues for (**e**) thorProfilin and (**f**) ΔN-thorProfilin. The chemical shift was fitted to the equation describing the equilibrium interaction of two biomolecules as described above. The $K_D$ estimated from this fits range between 200–500 μM.

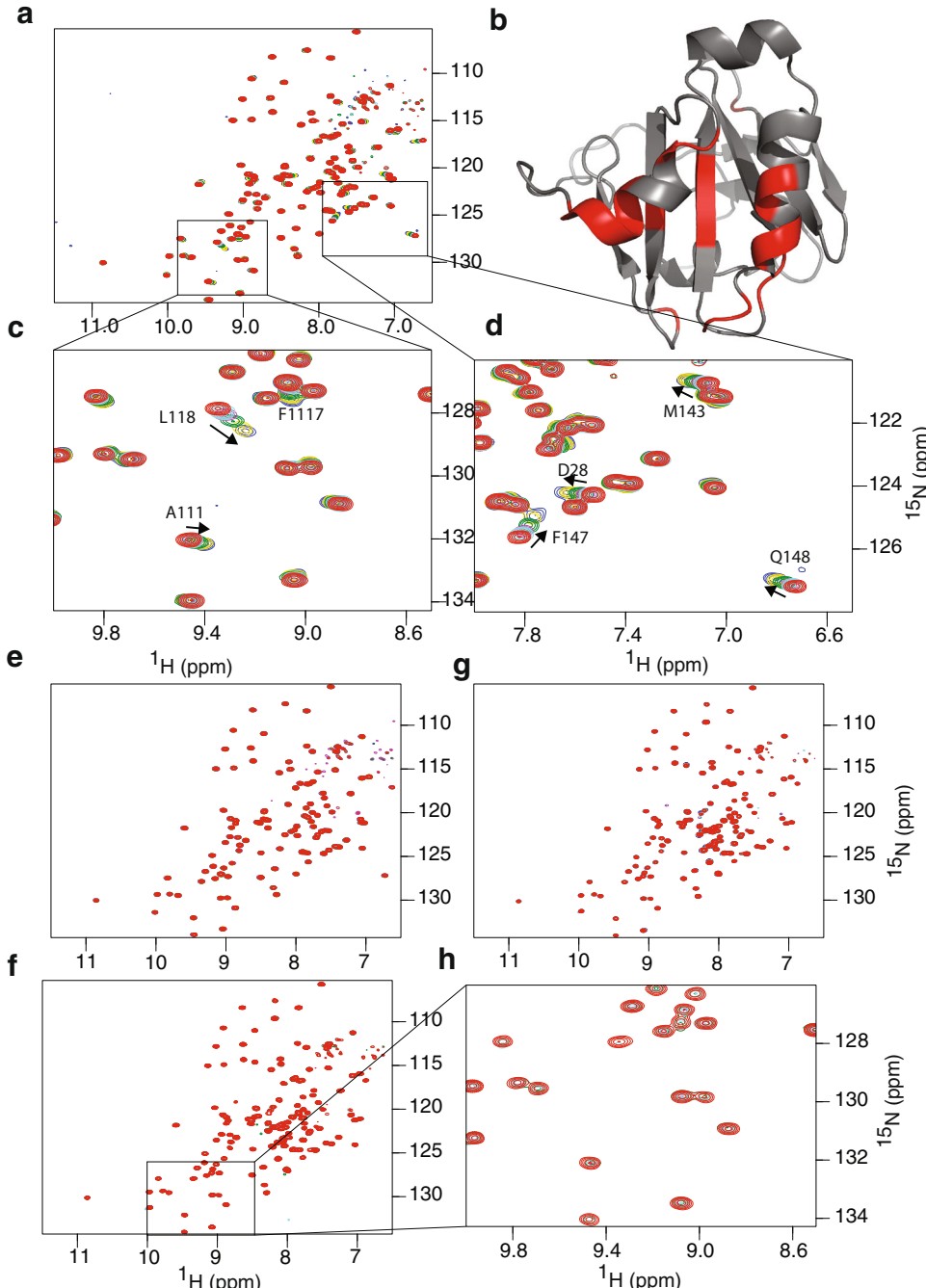

**Fig. 7 HeimProfilin N-terminal extension is important for interaction with phospholipids.** Phosphatidylinositol-4,5-bisphosphate (PtdIns-4,5-P$_2$ (PIP$_2$)) and D-myo-inositol-1,4,5-triphosphate (IP$_3$). **a** Overlay of $^1$H-$^{15}$N TROSY-HSQCs of ΔN-heimProfilin (400 μM) with increasing concentrations of PIP$_2$; 0 μM (red), 150 μM (cyan), 300 μM (magenta), 600 μM (green), 1200 μM (yellow) and 2100 μM (blue). **b** Structure of heimProfilin with the interacting residues color-coded. The interaction site appears to be located between the N-terminal and C-terminal helix. **c–d** Expansion of (**a**) showing a few residues belonging to the b-strand residues F117-W120 and residues Q110-A111 are also strongly involved in the interaction. **e** Overlay $^1$H-$^{15}$N TROSY-HSQCs of ΔN-heimProfilin (300 μM) with increasing concentrations of IP$_3$; 0 μM (red) 150 μM (cyan), 300 μM (magenta), 600 μM (green), 1200 μM (yellow), 2100 μM (blue). **f** Similar to (**a**) but with heimProfilin and PIP$_2$; 0 μM (red) 150 μM (cyan), 300 μM (magenta), 600 μM (green). **g** Similar to (**e**) but with heimProfilin interacting with IP$_3$; 0 μM (red) 150 μM (cyan), 300 μM (magenta), 600 μM (green), 1200 μM (yellow), 2100 μM (blue). Little or no chemical shift changes were observed for heimProfilin- IP$_3$ interaction. **h** Expansion of a region in (**f**).

profilins were able to inhibit actin polymerization in actin pyrene assays. Loki profilin was shown to have a distinctive loop (the Loki-loop) that was absent from the human profilin and that profilins from Loki, Odin, and Heimdall LC2 were able to interact with PIP$_2$ and inhibit their effect on actin polymerization. However, profilins from these Asgards were shown not to interact with polyproline motifs from VASP[5], whereas we show that both

the heimProfilin and the thorProfilin analyzed here do bind to polyproline motifs.

A small fraction of Asgard archaea (ca. 5%) encodes profilins with N-terminal extension, including the two profilins (heimProfilin and thorProfilin) analyzed in this contribution. Interestingly, deleting this extension enhanced binding to both polyproline motifs (for heimProfilin and thorProfilin) and PIP$_2$

(only tested for heimProfilin). These two molecules are important partners in the modulation of actin cytoskeleton dynamics in eukaryotes, but abrogates its direct modulation of the speed of filament elongation. These N-terminal extensions are however likely the result of independent events, possibly revealing convergent evolution. We cannot completely exclude that the extensions are the result of annotation errors. However, two lines of evidence strongly support that they are not artifacts: the general high reliability of start codon prediction of the annotation software, and the fact that the two extensions analyzed here have the same modulating effect on binding to polyproline. In addition, Asgards signatory proteins have been shown to display variable phylogenetic and domain distribution and this is proposed to result from dynamic evolution through horizontal gene transfer, domain shuffling, and gene duplication[21]. The extensions could thus play a role in modulating actin polymerization, presenting a novel regulatory mechanism to the best of our knowledge unique to some members of the Asgard archaea, indicating that polyproline-mediated regulation could predate the Asgard-Eukarya split.

We propose a model where modification of the extended N-terminal or interaction with third-party proteins and molecules, causes heimProfilin to behave similarly to ΔN-heimProfilin. This will allow for PIP$_2$ and polyproline interactions. Concurring or subsequent remodification would flip heimProfilin to an actin modulating state, allowing for actin polymerization regulation. Modification or interaction with third-party proteins would then be able to reset profilin to the first step of the cycle.

In the unlikely case where the N-terminal extensions are artifacts due to annotations errors, the regulation model proposed above would not be valid anymore. On the other hand, it would imply that heimProfilin binds to both polyproline motifs and PIP$_2$, and that thorProfilin binds to the former, generally supporting the conclusion of the next paragraph.

In light of these results and with the low affinities observed between the interaction of Asgard archaea profilins and polyproline motif it is tempting to say that most Asgard archaea profilin might interact with polyprolines motifs including those from previously studied of Loki-1 and 2 and heimdall LC2 archaea.

In conclusion, this study suggests that Asgard archaea encode a complex cytoskeleton functionally analogous to major eukaryotic cytoskeletal characteristics. Moreover, Heimdallarchaeota LC3 (and most likely many other archaea) expresses profilins that are potentially regulated by phospholipid binding and polyproline interaction, something which was long thought to be eukaryotic-specific, and previously not observed in other Asgard archaea.

## Methods

### Protein expression and purification.
The profilins (sequence ID OLS22855.1-Heimdallarchaeota LC3[8], sequence ID TFG12995.1- Thorarchaeota) and actin (sequence ID OLS30618.1-Heimdallarchaeota AB_125[8] and TET76256.1-Heimdallarchaeota_E44_bin5[22], respectively) used in this study were from Heimdallarchaeota and Thorarchaeota. Sequence OLS30618.1 has a 35 amino acids C-terminal deletion compared with TET76256.1. Thus the OLS30618.1 variant is from now on named ΔC-heimActin and the TET76256.1 heimActin. Heimdallarchaeota genes were sub-cloned into the pSUMO-YHRC vector (kindly provided by Claes Andréasson (Addgene Plasmid #54336; RRID: Addgene_54336)) with an N-terminal 6xHistidine-tag and a SUMO-tag (cleavable with Ulp1 protease). Thorarchaeota profilin genes were ordered from geneScript in a pET28a vector with an N-terminus 6xHistidine-lipolyl tag with a TEV protease cleavage site. Recombinant proteins were overexpressed in E. coli Rosetta (DE3) strain or BL21* (Thorarchaeota and heimProfilin mutants). HeimProfilin mutants were expressed with C-terminal 6xHistine tagged cloned in the pET21a vector. Initially, the cells were grown at 37 °C in 2x TY broth. Protein expression was induced with 0.5 mM IPTG when the optical density at A$_{600}$ was 0.6–0.8. After induction, the cells were grown overnight at 30 °C. For actin expression, the cells were grown for 4 h at 20 °C post-induction. The cells were harvested by centrifugation and the cell pellet was dissolved in the binding buffer; 50 mM Tris-HCl pH 8.0 or pH 7.5 (for heimProfilin and thorProfilin), 0.3 M NaCl/KCl, 1 mM TCEP, 10 mM imidazole, 10% glycerol. For actin, the buffer was supplemented with 2 mM MgCl$_2$.

### Protein expression for NMR spectroscopy.
The pSUMO-YHRC vector with the heimProfilin (sequence ID OLS22855.1) and ΔN-heimProfilin (with the first 1–23 amino acids in the N-terminal extension deleted from the sequence OLS22855.1) was transformed and expressed in E. coli Rosetta DE 3 cells. The cells were grown in 2x TY media at 37 °C until A$_{600}$ = 0.8. The cells were harvested by centrifugation at 4000 × g for 15 min and washed twice with M9 medium. The cells were thereafter grown overnight at 30 °C in M9 medium, supplemented with 1 g/L $^{15}$N-ammonium chloride and 1 g/L $^{13}$C-glucose. Protein expression was induced with 0.5 mM IPTG. For the labeling of protein with Deuterium ($^2$H), the M9 medium was prepared in D$_2$O. The cells were harvested by centrifugation and the cell pellet was dissolved in the binding buffer; 50 mM Tris-HCl pH 7.5, 0.3 M NaCl, 1 mM TCEP, 10 mM imidazole, 10% glycerol. Cells were lysed by sonication, followed by centrifugation at 25,000 × g for 45 min. The supernatant was loaded onto a His-GraviTrap column (1 mL, GE Healthcare) pre-equilibrated with binding buffer. The 6xHistidine tagged proteins bound to the column were eluted with a binding buffer containing 250 mM imidazole. The eluted proteins were incubated with ULP1 protease overnight at 4 °C for tag cleavage, followed by buffer exchange on a PD-10 column (GE Healthcare). The tag was removed by reloading the protein solution onto the His-GraviTrap column. The proteins were concentrated using a 10,000 NMWL cutoff centrifugal filter (Merck-Millipore). The concentrated proteins were subjected to size-exclusion chromatography on Superdex-200 or Superdex-75, 10/300 GL column (GE healthcare) as the final purification step. For size-exclusion chromatography, the column was pre-equilibrated with 25 mM Tris-HCl pH 8.0 (or 7.5 for profilin), 50 mM NaCl, 1 mM TCEP, 1 mM MgCl$_2$ (for actin only) and 10% glycerol. Fractions containing the purified proteins were pooled, concentrated, and stored at −80 °C for further use. The pET28a vector with His-lipolyl-thorProfilin with cysteine at position 2 mutated to alanine (sequence ID TF12995.1) and His-lipolyl-ΔN-thorProfilin (with the first 1–18 amino acids in the N-terminal extension deleted from the sequence TF12995.1) was transformed and expressed in E. coli BL21*. The cells were grown in 2x TY media at 37 °C until A$_{600}$ = 0.8. The cells were harvested by centrifugation at 4000 × g for 15 min and washed twice with M9 medium. The cells were thereafter grown overnight at 30 °C in M9 medium, supplemented with 1 g/L $^{15}$N-ammonium chloride. Protein expression was induced with 1 mM IPTG. The cells were harvested by centrifugation and the cell pellet was dissolved in the binding buffer; 50 mM Tris-HCl pH 7.5, 0.3 M NaCl, 10 mM imidazole, 1% Triton X100. Cells were lysed by sonication, followed by centrifugation at 45,000 × g for 40 min. The supernatant was loaded onto Nickel charged Sepharose column pre-equilibrated with binding buffer. After wash, the bound proteins were eluted with a buffer containing 500 mM imidazole. The eluted proteins were desalted using a PD10 column (GE Healthcare) and incubated with TEV protease overnight at room temperature for tag cleavage. The tag was removed by reloading the protein solution onto the Nickel charged column. The proteins were concentrated using a 5000 NMWL cutoff centrifugal filter (Merck-Millipore). The concentrated proteins were subjected to size-exclusion chromatography on a Superdex-75 GL column (GE healthcare) as the final purification step. For size-exclusion chromatography, the column was pre-equilibrated with 25 mM Tris-HCl 6.8, 150 mM NaCl. Fractions containing the purified proteins were pooled, concentrated, and stored at −20 °C for further use. Protein identity was confirmed by mass spectrometry.

### Nuclear magnetic resonance experiments.
All NMR experiments were done on Bruker spectrometers equipped with triple resonance cryogenic probes operating at proton larmor frequencies of 600, 700, and 800 MHz. Experiments for the assignment were as previously reported[13]. All NMR binding titrations were done on the Bruker 600 MHz at 298 K unless otherwise stated. For heimProfilins and mutant, all protein samples were either single labeled $^{15}$N, or double-labeled $^{15}$N, $^{13}$C, at concentrations between 5–10 mg/mL in 25 mM Tris-HCl pH 7.5, 50 mM NaCl, 5% glycerol and supplemented with 3% D$_2$O, and 0.03% sodium azide. 2D $^1$H-$^{15}$N TROSY-HSQC experiments were recorded at a fixed amount of heimProfilin or ΔN-heimProfilin (200–400 μM) with increasing amounts of heimActin, ΔC-heimActin, polyproline (PPPAPPLPAAQ), or phospholipids (Phosphatidylinositol-4,5-bisphosphate (PtdIns (4,5)P$_2$ (PIP$_2$) of D-myo-inositol-1,4,5-triphosphate (IP$_3$)). Experiments involving heimActin or ΔC-heimActin were done in the presence of 0.9 mM latrunculin to inhibit polymerization. T$_2$ times were measured for heimProfilin, ΔN-heimProfilin free and bound to polyproline. For these experiments, a similar pulse program and parameter set as described in[13] was used with the only exception that the relaxation delay was increased to 2 s. For thorProfilin, protein samples were labeled with $^{15}$N, 5 mg/mL in 25 mM Tris-HCl pH 6.8, 150 mM NaCl and supplemented with 3% D$_2$O and 0.03% sodium azide. 2D $^1$H-$^{15}$N TROSY-HSQC experiments were recorded at a fixed amount of thorProfilin or ΔN-thorProfilin (200 μM) with increasing amounts of polyproline (PPPAPPLPAAQ). All experiments were processed with Bruker TopSin software and analyzed with the CcpNmr analysis program[23] and Bruker DynamicCenter2.5.3.

### Structure determination.
The assignments of Heimdallarchaeota profilin used in this study were from reference Haq et al.[13]. In addition, we measured 3D $^1$H–$^1$H NOESY resolved in $^{13}$C–$^1$H and $^1$H–$^{15}$N TROSY experiments with the following specifications: 80 ms mixing time and 128 ($^{15}$N or $^{13}$C) × 256 ($^1$H) × 2048 ($^1$H, direct) were measured and used for distant restraint determination. We also used the $^3J_{HNH}\alpha$ couplings and together with the NOE derived distances were deposited

## Table 1 NMR and refinement statistics for Heimdallarchaeota profilin.

| | Heimdallarchaeota profilin |
|---|---|
| NMR distance and dihedral constraints | 1980 |
| **Distance constraints** | |
| Total NOE | 1135 |
| Intra-residue | 161 |
| **Inter-residue** | |
| Sequential ($|i - j| = 1$) | 322 |
| Medium-range ($|i - j| < 4$) | 252 |
| Long-range ($|i - j| > 5$) | 400 |
| **Intermolecular** | |
| Hydrogen bonds | 43 |
| **Total dihedral angle restraints** | |
| $\phi$ | 98 |
| $\psi$ | 98 |
| $^3J_{HN\alpha}$ scalar couplings | 83 |
| **Structure statistics** | |
| Violations (mean and s.d.) | 0 |
| Distance constraints (Å) | 0.3 |
| Dihedral angle constraints (°) | 0 |
| Max. dihedral angle violation (°) | 0 |
| Max. distance constraint violation (Å) | 0.3 |
| **Deviations from idealized geometry** | |
| Bond lengths (Å) | 3 |
| Bond angles (°) | 0 |
| Impropers (°) | 0 |
| **Average pairwise r.m.s. deviation (residues 21-145 (Å)** | |
| Heavy | 1.05 ± 0.27 |
| Backbone | 0.50 ± 0.22 |

"Pairwise r.m.s. the deviation was calculated among 20 refined structures."
Ramachandran statistics are in the Methods section at end of the Refinement subsection.

in BMRB with ID: 50190. Structure calculations were done using the CYANA 3.98.13[24] package in two steps. First, the NOESY cross-peaks were converted into upper distance restraints in an automated process in CYANA. The $\phi/\psi$ dihedral angles were determined from backbone chemical shifts using TALOS-N[25] and together with $^3J_{JHNH}\alpha$ were used as input for the structure calculations. The structures were calculated with 200,000 torsion angle dynamics steps for 100 conformers starting from random torsion angles by simulated annealing. The resultant structures were further refined and energy minimized in explicit water using cartesian dynamics within CNS 1.2[26–28]. Ramachandran statistics are as follows: most favored regions, additionally allowed regions, generously allowed regions, and disallowed regions 66, 32.5, 1.3, and 0.2% respectively. For representation and analysis, the 20 conformers with the lowest energy and no significant violations of the experimental data values were selected. The structural statistics together with all input data for the structure calculations are presented in Table 1. The structural coordinates have been deposited in the protein data bank with PDB ID: 6YRR.

**Electron Microscopy.** For EM observation, heimActin and ΔC-heimActin (5 μM) were polymerized for 2 h and ultracentrifuged for 1 h at 150,000 × g at 4 °C. The pellets were dissolved in F-actin buffer (20 mM Tris-HCl pH 8.0, 200 mM KCl, 1 mM ATP, 4 mM MgCl₂) and applied to carbon-coated grids for 60 s and negatively stained with 1% (w/v) uranyl acetate. A TECNAI G2 spirit Bio-TWIN electron microscope (FEI Company) was used at an accelerating voltage of 80 kV with a 70 μm objective aperture and a 100 μm condenser aperture at a nominal magnification of 1.8–3.0 × 104.

**TIRF–time-resolved microscopy.** The experiments were carried out on a custom-built prism-based TIRF microscope[29] using PEG (poly[ethylene glycol])-coated quartz slides functionalized with biotin[30]. The slides were first incubated with streptavidin for 5 min (200 ug/ml streptavidin from Thermo, 10 mM Tris-HCl pH7.5 and 50 mM KCl) and then blocked 0.3 mg/L of BSA. Unlabeled rabbit actin was mixed with Alexa Fluor -568 labeled actin (0.5%) and Biotin-actin (0.3%) at a final concentration of 2 μM in TIRF-buffer (20 mM imidazole, pH 7.4, 50 mM KCl, 1 mM MgCl₂, 1 mM EGTA, 0.2 mM ATP, 15 mM glucose, 20 mM β-mercaptoethanol and 0.5% w Tris-HCl pH 7.5, 0.5% methylcellulose, supplemented with 1 μl oxygen scavenging system (1.25 mg/ml glucose-oxidase, 0.2 mg/ml catalase). Actin mix was kept on ice before being polymerized by adding 10× polymerization buffer (1 M KCl, 10 mM MgCl₂) supplemented with heimProfilin, ΔN-heimProfilin, L134A, G104A, G63A, and F95A (100 μM). This solution, 120 μl was briefly incubated at 37 ºC for 5 min before gently flowed into the microscope

chamber. Images of actin filaments were collected continuously after the introduction of the polymerization mixture into the flow cell with 500 ms exposure. Alexa 568 fluorescence was excited with a 532 nm Nd: YAG laser. Data acquisition was controlled using MicroManager[31].

**ATPase measurements.** Purified heimActin was diluted to a final concentration of 1 μM using the following actin polymerization buffer (20 mM Tris-HCl 7.5, 50 mM KCl, 2 mM MgCl₂, 1 mM ATP). The reaction was performed at different temperatures (20–70 °C) for 30 min. Phosphate production was measured using a PᵢColorLock Gold Colorimetric Assay kit (Innova Biosciences) according to the manufacturer's instructions. Absorbance was measured at 635 nm on a Perki-nElmer EnSpire microplate reader.

**Isothermal titration calorimetry.** ITC experiments were performed on an ITC200 system (MicroCal). Samples were first dialyzed against 50 mM Na₃PO₄ pH 7.5, 50 mM NaCl. The purified recombinant protein (280 μl, approx. 0.1–0.5 mM) was placed in the cell and titrated with 20 injections of 0.4 μl of ligand (1.0–14 mM polyproline (VASP)) with 1–2 min between each injection while stirring at 1500 RPM. We used two different versions of the VASP peptide; a long (GAGGGPP-PAPPLPAAQ) and a short version (PPPAPPLPAAQ). Experiments were done in triplicates at 25 °C. Data analyses were carried out using Origin 5.0 (MicroCal) provided by the manufacturer.

**Fluorescence of pyrene–actin interaction.** Pyrene-rabbit actin was subjected to size-exclusion chromatography to obtain the monomeric fraction. Pyrene-labeled rabbit actin (Cytoskeleton, Inc.) polymerization assay was performed in a 96-well, black, flat bottom plate (Corning, Nunc). The pyrene-rabbit actin (2%) in 2 mM Tris-HCl pH 7.5, 0.5 mM ATP, 0.5 mM DTT, 0.2 mM CaCl₂, alone or with 0.3 μM actin seed in the presence of different concentration of heimProfilin or ΔN-heimProfilin (50–280 μM) were used. The reaction was initiated with the addition of a 10x polymerization buffer (1 M KCl and 10 mM MgCl₂). The increase in fluorescence intensity due to polymerization was measured at the excitation and emission wavelength of 365 and 410 nm, respectively using the Fluoroskan Ascent FL spectrofluorometer (Thermo Scientific). The fluorescence intensity (a.u) was then plotted against time. The initial slope was estimated thus; fluorescence intensity resulting from the lag phase was omitted, then a linear slope was fitted to the fluorescence intensity corresponding to the linear phase of the reaction. These slopes were then plotted against heimProfilin or ΔN-heimProfilin concentration. These plots were done with the KaleidaGraph software.

**Sedimentation assay.** Actin polymerization and its binding to profilin were determined by sedimentation assays. Two different buffers were used. Polymerization buffer 1: For ΔC-heimActin (2 mM Tris-HCl pH 7.5, 100 mM KCl, 1 mM ATP, 2 mM MgCl₂, 10 mM imidazole) and Polymerization buffer 3: For heimActin (20 mM Tris-HCl pH 8.0, 250 mM KCl, 1 mM ATP, 4 mM MgCl₂). These buffers were chosen as they promote a higher content of polymerized actin. Polymerization was initiated by diluting actin (5 μM) with 10x polymerization buffer and incubating for 2 h at room temperature. The polymerized actin filaments were pelleted by ultracentrifugation at 150,000 × g for 1 h, at 4 °C using the TLA-55 rotor Optima MAX-XP ultracentrifuge (Beckman-Coulter). The pellet was carefully separated and resuspended in the same volume as the supernatant and analyzed by SDS-PAGE. To measure the effect of heimProfilin on heimActin polymerization dynamics, 5 μM heimActin were mixed with different concentrations (0, 1, 5, 10, and 20 μM) of heimProfilin or ΔN-heimProfilin and diluted with 10x buffer to initiate the polymerization. The reaction mixture was incubated for 2 h at room temperature and the filamentous actin was pelleted by ultracentrifugation at 150,000 × g for 1 h.

**Analytical size exclusion chromatography.** Molecular weight markers (Gel Filtration Calibration Kit HMW, from Cytiva, Formerly, GE Healthcare) were prepared at a concentration of 1 mg/mL by dissolving in analytical HPLC buffer (25 mM Tris-HCl pH 7.0, 50 mM KCl, 0.2 mM ATP, 0.2 mM CaCl₂, 1 mM DTT and 5% glycerol). Analytical HPLC was performed on Simadzu LC-20AT at 4 °C fitted with Yarra 3 μM SEC 3000, 150 × 7.8 mm column with the buffer solution as mobile phase, at a flow rate of 0.5 mL/min, with a sample injection volume of 100 μL. Protein samples were either dissolved in HPLC buffer alone or in the presence of 100 mM DTT.

**Phylogenetic analysis.** Sequences homologous to the heimProfilin (OLS22855.1) from 'Candidatus Heimdallarchaeota archaeon' LC_3 were gathered from the nr database at NCBI using PSI-BLAST[32], with an E-value threshold of 1e−4. After six iterations, no further hits were retrieved from the Asgard archaea, and the first hits were found in Eukaryotes. All proteins (290, including 27 eukaryotic profilins) were downloaded and aligned with MAFFT[33], using the L-INS-i method. The alignment was trimmed with trimAl, using the -gappyout option, resulting in a 118-aminoacid long alignment[34]. A maximum-likelihood phylogenetic tree was inferred with IQ-TREE 2[35], with automatic model testing and drawing 1000 ultra-fast bootstraps with UFBoot2[36]. The model with the best fit according to the Bayesian

Information Criterion was LG + F + R6, i.e. the LG matrix, counting frequencies from alignment (F) and using a FreeRate model with 6 categories (R6). The alignment was visualized and analyzed with SeaView 4[37].

**Statistics and reproducibility**. TIRFM data analysis was performed by manual filament tracking with the *segmented line tool*, followed by *Create kymograph* using multi kymograph and a line thickness of 7 pixels, from FIJI[38]. Slopes from kymographs were measured to determine the elongation rate of individual filaments. The pixel size/length was converted into microns/s and monomers/s. One actin monomer was assumed to contribute 2.7 nm of length to the actin filament[39]. For the presentation of data 21 frames, representing 10.5 s, of a growing filament was extracted, followed by subtraction of the median of the 6 preceding frames from each of the 21 frames. Finally, a $3 \times 3$ median filter was applied to the resulting stack to reduce noise. Data were processed and figures were generated in R (v.4.0.5), using R-Studio and ggplot2[40]. Filaments that suddenly appeared, i.e. have elongation rates much, much higher than previously published[39] were likely pre-polymerized and just binding to the glass, and were therefore discarded. Growth velocity was determined by averaging the growth of 6–13 visible time-lapse filaments. The variation of the speeds is represented with the spread of the box plot.

The ATP analysis was a replicate of four independent measurements. The replicate here represents experiments repeated four times on four occasions with different sample preparations.

**Reporting summary**. Further information on research design is available in the Nature Research Reporting Summary linked to this article.

## Data availability
The structure reported here has been deposited in the Protein Data Bank with accession number 6YRR. The raw data, figures, and material are all available in Supplementary Information including supplementary movie 1 or from the corresponding authors on reasonable request.

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

## Acknowledgements
We would like to thank Dr. Mats Pettersson from Uppsala University for his helpful contribution in the writing the R scripts used in the analysis. This work was supported by Wenner-Gren Stiftelsen Fellow's Grants, Ake Wiberg, Magnus Bergvall, and O.E Edla Johannsson foundation grants to C.C, Swedish Research Council Grant 621-2013-4685 for F.H. and Wellcome Trust Grant 203276/F/16/Z for S.R.H., S.S. and F.H. This study made use of the NMR Uppsala infrastructure, which is funded by the Department of Chemistry - BMC and the Disciplinary Domain of Medicine and Pharmacy as well as the Imaging facility at Stockholm University.

## Author contributions
S.S., F.H., S.R.H., A.-C.L. and C.C. conceived the project. C.C., S.S., S.R. H. and F.H. cloned, expressed, and purified all proteins. S.S. performed the pyrene and sedimentation assay, and electron microscopy. F.H. performed the ATP assay. C.C. performed all NMR and ITC experiments and structure calculation. K.J.R. performed structure calculation refinements and minimizations. C. C. and J. E. performed TIRF microscopy and J.E.

analyzed the TIRF data. L.G. performed the evolutionary analysis of the profilin homologs. C.C. wrote the paper with contributions from all other authors.

## Funding

## Competing interests

The authors declare no competing interests.
