## [Peer Review File · Communications Biology]

Reviewers' comments:

Reviewer #1 (Remarks to the Author):

This paper describes the characterization of proteins of Heimdallarchaeota LC3, a member of the Asgard super phylum which are thought to constitute a link between eukaryotic cells and its precursors. These proteins show some "eukaryotic" behavior which questions the commonly understood sequence of events during evolution. Specifically, heim-Profilin, heim-Actin, and their interactions with polyproline and phospholipids are investigated. HeimProfilin shows structural differences to both human and Loki (another superphylum). However, there are also similarities with human Profilin in terms of functions (actin polymerization and interaction with polyproline and phospholipids that regulate this interaction).

This paper presents a very relevant step in understanding evolutionary origin of eukaryotes, and when I was reading it I grow curious and couldn't put it down. Therefore I recommend publication of this work.

The following points have to be addressed:

- The presentation and interpretation of the interactions of heimProfilin and PIP2 is somewhat confusing and contradictory. For example line 209: Fig. 6 (indicated in the phrase) does not show what is mentioned in the phrase, it's actually in Supp Fig 11. Further, it is said that heimProfilin does not interact with PIP2, but then a Kd is determined - that is a contradiction. Then again the authors list which residues interact, etc. I suggest to stick to one description, for example "interacts weakly", so that all observation that follow are in line with it.
- The authors show the solved structure of heimProfilin in Fig 1: show the position 72-75, where the helix is missing in heimProfilin.
- DeltaN-profilin has higher affinity to VASP than normal profilin. That does not make any sense at first sight. This would actually tell me that Heimdallarchaeota LC3 does NOT have the characteristics of eukaryotes. On the other hand, from the introduction and abstract we already know that the authors claim that it actually HAS those characteristics. Finally, this contradiction is resolved in the discussion, where the authors propose a model where additional proteins suspend the inhibitory effect of the N-expansion. I leave it up to the authors, but I suggest to add a reason to the introduction why both the N-truncated version and the wild type of profilin are used and pre-inform the reader that it is not a contradiction that DeltaN-profilin is the active version. Then, the reader would not build up a sense of confusion in the course of reading the manuscript.
- Line 215/216. I don't understand the statement "slower than internal correlation time". If the pulse sequence is designed to detect motion slower than 5 ms (through appropriate refocusing pulse spacing) it will detect internal motion slower than 5 ms, and not faster motion. The way it is written, it implies that the authors have already known on what timescale internal motions are present.

Minor:

- line 72: delete "and"
- line 82: "more dynamic" relative to what? I assume relative to the folded residues.
- line 121: incomplete phrase
- line 135: Should be Figs 3f-g instead of 2f-g
- line 146: "were" should be "where"
- line 237: "General" not capital
- line 356 "delayed"
- line 369: "they" missing

- line 373: distances cannot be obtained from CS and TALOS. They mean to say that phi/psi are obtained from CS. Rephrase.
- line 383: "centrifuged"
- Fig 6: Second phrase is incomplete.
- Supp Fig 5: correct "normalized"
- Supp Fig 7: labeling a-e seems messed up
- Supp Fig 9: "sites" instead of "sides"
- Supp Fig 12: c) is expansion of b), not a); "submillimolar range" probably is meant to be "sub millisecond range"
- references 12 and 21 are identical

Reviewer #2 (Remarks to the Author):

Review of Survery et al "Heimdallarchaea encodes profilin with eukaryotic-like actin regulation and polyproline binding"

This paper includes structural and biochemical studies of the activity of a heterologously produced Profilin protein, whose sequence was obtained from the deep branching Heimdallarchaeota candidate phylum, with actins from these species and rabbit. The results suggest that the Heim profilin regulates polymerization of Heim actin, which are novel findings. There is some evidence presented that it weakly affects rabbit actin too, though it is not clear how specific or useful these latter results are. There is also doubt about the overall approach of using eukaryotic reagents (proteins/peptides/lipids) in ongoing binding or biochemical studies with highly divergent species. What is the aim or relevance? Evolutionary analyses based on detailed sequence or 3D structure analysis are not subject to the frequently spurious results obtained by combining molecules separated by many millions of years of evolution. The data itself is interpretable although not very thorough in several locations (clearer information on replication would be useful). The presentation is quite nice for some of the figures, although difficult to follow at places and by improving the annotation and clarity of figures the paper would be significantly easier to follow. I hope the further critique points below can help the authors develop the manuscript.

The NMR structure appears poor and needs refinement (many Ramachandran outliers) and/or a proper explanation of the data. Some of the structural features described are quite subtle and the structure may not be accurate enough to have confidence in them.

There are 3D structural and sequence-length differences noted between regions of lokiarchaeota- and Heim-profilins, but the data shown does not demonstrate whether these are conserved and characteristic differences between the candidate phyla profilins, as implied, or whether there is significant variability within each taxon. For example, showing well-analyzed alignments, from a large, diverse and balanced sequence set, and corresponding molecular phylogeny of the profilins from across Asgard organisms and eukaryotes would enable the conservation of these features to be identified – these are mentioned later in the manuscript, but data are not shown, apart from an unclearly annotated alignment Supp fig 11.. Even if sequence is poorly conserved, the groups should still be resolvable and, particularly, the conserved differences in length of the various regions should be detectable. Are there particular conserved residues that appear, which contribute to the structural differences noted? Are these features predicted to be conserved in the ribosomal trees of the phyla? What is the conservation of length and identity of residues in the N-terminal extension? These would substantiate the more general claims being made about conservation of differences amongst the profilin groups in the major Asgard taxa.

Ln 97. And 101. The data in Fig2C/G appear indistinguishable from normal actin (2a/f) visually, but

these sentences suggest a slight difference. Clarify description.

Ln 100. Fig 2f/g The activity on rabbit actin appears very low compared to HeimActin, as might be expected. This should be admitted/noted. How does it generally compare to Loki or other heterologous profilins?

Ln 103. Why was the AB_125 actin (fragment) used? As it is not a complete sequence, if its remaining sequence differs from LC3, it cannot be compared properly as a deletion of the C-term (Ln 112). Also, what would be the purpose of deleting the C-terminal 35 amino acids? The way AB_125 protein data is interpreted/discussed should be significantly revised.

Ln 106. Were the SDS-PAGE conditions reducing or non-reducing? Clarify these conditions in the relevant figure legends and methods, and why they were chosen. In sufficient data are included to determine whether the protein is running as a monomer, dimer or other multimer under the various conditions.

Ln 111. Supp Fig 5 data are not well analyzed and the molecular weight claim for heimActin (44 kDa) is weak. Interpolation from a fit to a standard curve for peak maxima of marker proteins is needed as a basic analysis of the supp fig 5 data. Here, also provide the reader with the calculated MW from the sequence, as a comparison. But MW cannot be strictly confirmed by SEC, which is also subject to shape effects and calibration requirements like SDS-PAGE. It is important to show reducing versus non-reducing SDS-PAGE or SEC for proper comparisons to be made. MW can be determined by SEC-MALS or ES-MS, or other more direct methods.

pp4-5, and pp6-8 Split-up the two enormous paragraphs, that make the points hard to follow.

Ln 117. The profilins do not necessarily appear 'efficient' based on the presented data. Quite a moderate effect is seen in SuppFig4a-b, j-m. (also, please note the actin concentration somewhere here for reference). Given this moderate effect, the experimental variance (errors) should ideally be included in quantified data based on identical replicate polymerization experiments (in panels j-m). Here I also found the bar graphs inappropriate for the type of data – some sort of scatter or line plot would be much clearer, with y-axis as percent of total protein. If showing only the gels or single quantification for a non-quantified basic test, it would be appropriate to include higher concentrations of profilin, to show (presumably) a more complete retention of actin in the supernatant.

Ln 126 Sup Fig 4d-e The results indicate that there is no detectable influence of these profilins on rabbit actin polymerization (pelleting). The claim that these profilins interact with monomeric actin appears very weak and would require a stronger case.

Ln 135 Supp Fig 6/7. HSQC changes look minimal upon titration of actin and do not appear consistent with specific binding to HeimActin. Is there titration data with different concentrations of HeimActin to show the chemical shift transitions in the putative intermediate binding states?

Ln 145. These lists of interacting residues are not helpful in comparing profilin to eukaryotic actin. It would need a figure, and the residues hint they could be very different. This observation should not be minimized or hidden – Supp Fig 6 does not compare these two as this section suggests (Ln 148) – revise to avoid the mistaken implication.

Ln 147. Were >> Where

Ln 151 Are polyproline regions found in Heimdallarchaeota?

Ln 157. (fig 3) These interactions are weak and, more concerning, they don't appear to have a defined stoichiometry (the binding model is not clear). The native protein has a significantly weaker affinity

than the truncated, and the chemical shift perturbations are also weak in Supp Fig 9 and show no particular concentration of shifts to one region – furthermore, they do not show why the native and truncated proteins would differ. These findings suggest binding is non-specific, or non-physiologically relevant, so relevance of these are in doubt.

Ln 193 is not enough information or data provided on ThorProfilin to determine any features of binding to polypro or its significance. (Fig 5). The role of the ThorProfilin N-terminal extension, where a difference was observed for Heimprofilin, remains unclear and appears inconsistent with the Heim

Ln 194 Thro >> Thor

Ln 207. Does the lack of binding to the apparent full-length protein with polypro and PIP indicate this region is not expressed in the archaeon, or that these putative interactions are not physiologically relevant? This section was confusing – what interaction is tighter, given previous sentence said there was no binding (Ln 211)?

Ln 227. The clustering of shifted residues indicates 'a potential binding interface for PIP2'. Would such clustering imply that there is no such interface for the previous PolyPro interaction that show limited or no clustering? Are the clusters this located in a homologous region to PIP binding to other profilins?

Reviewers' comments:

Reviewer #1 (Remarks to the Author):

This paper describes the characterization of proteins of Heimdallarchaeota LC3, a member of the Asgard super phylum which are thought to constitute a link between eukaryotic cells and its precursors. These proteins show some "eukaryotic" behavior which questions the commonly understood sequence of events during evolution. Specifically, heim-Profilin, heim-Actin, and their interactions with polyproline and phospholipids are investigated. HeimProfilin shows structural differences to both human and Loki (another superphylum). However, there are also similarities with human Profilin in terms of functions (actin polymerization and interaction with polyproline and phospholipids that regulate this interaction).

This paper presents a very relevant step in understanding evolutionary origin of eukaryotes, and when I was reading it I grow curious and couldn't put it down.

Therefore I recommend publication of this work.

The following points have to be addressed:

- The presentation and interpretation of the interactions of heimProfilin and PIP2 is somewhat confusing and contradictionary. For example line 209: Fig. 6 (indicated in the phrase) does not show what is mentioned in the phrase, it's actually in Supp Fig 11. Further, it is said that heimProfilin does not interact with PIP2, but then a Kd is determined - that is a contradiction. Then again the authors list which residues interact, etc. I suggest to stick to one description, for example "interacts weakly", so that all observation that follow are in line with it.

We agree with the reviewer that this part might feel confusing. This confusion was possibly due the fact that the referencing to the figures was not clear. We have added more description to the sentences and further referenced them more clearly.

- The authors show the solved structure of heimProfilin in Fig 1: show the position 72-75, where the helix is missing in heimProfilin.

- DeltaN-profilin has higher affinity to VASP than normal profilin. That does not make any sense at first sight. This would actually tell me that Heimdallarchaeota LC3 does NOT have the characteristics of eukaryotes. On the other hand, from the introduction and abstract we already know that the authors claim that it actually HAS those characteristics. Finally, this contradiction is resolved in the discussion, where the

authors propose a model where additional proteins suspend the inhibitory effect of the N-expansion. I leave it up to the authors, but I suggest to add a reason to the introduction why both the N-truncated version and the wild type of profilin are used and pre-inform the reader that it is not a contradiction that DeltaN-profilin is the active version. Then, the reader would not build up a sense of confusion in the course of reading the manuscript.

We have reformatted and further incorporated the findings early on in the manuscript to clarify this point.

- Line 215/216. I don't understand the statement "slower than internal correlation time". If the pulse sequence is designed to detect motion slower than 5 ms (through appropriate refocusing pulse spacing) it will detect internal motion slower than 5 ms, and not faster motion. The way it is written, it implies that the authors have already known on what timescale internal motions are present.

We agree that this statement was unclear and have corrected the sentence.

Minor:

- line 72: delete "and" **corrected.**
- line 82: "more dynamic" relative to what? I assume relative to the folded residues. **corrected to "relative to the rest of the of the protein chain".**
- line 121: incomplete phrase **corrected.**
- line 135: Should be Figs 3f-g instead of 2f-g **corrected.**
- line 146: "were" should be "where" **corrected.**
- line 237: "General" not capital **corrected.**
- line 356 "delayed" **corrected.**
- line 369: "they" missing. **corrected.**
- line 373: distances cannot be obtained from CS and TALOS. They mean to say that phi/psi are obtained from CS. Rephrase. **Rephrased.**
- line 383: "centrifuged" **corrected.**
- Fig 6: Second phrase is incomplete. **Corrected.**
- Supp Fig 5: correct "normalized" **corrected.**
- Supp Fig 7: labeling a-e seems messed up. **Corrected.**
- Supp Fig 9: "sites" instead of "sides" **corrected.**
- Supp Fig 12: c) is expansion of b), not a); "submillimolar range" probably is meant to be "sub millisecond range" **corrected.**

- references 12 and 21 are identical **corrected**.

Reviewer #4 (Remarks to the Author):

Review of Survery et al "Heimdallarchaea encodes profilin with eukaryotic-like actin regulation and polyproline binding"

This paper includes structural and biochemical studies of the activity of a heterologously produced Profilin protein, whose sequence was obtained from the deep branching Heimdallarchaeota candidate phylum, with actins from these species and rabbit. The results suggest that the Heim profilin regulates polymerization of Heim actin, which are novel findings. There is some evidence presented that it weakly affects rabbit actin too, though it is not clear how specific or useful these latter results are. There is also doubt about the overall approach of using eukaryotic reagents (proteins/peptides/lipids) in ongoing binding or biochemical studies with highly divergent species. What is the aim or relevance? Evolutionary analyses based on detailed sequence or 3D structure analysis are not subject to the frequently spurious results obtained by combining molecules separated by many millions of years of evolution. The data itself is interpretable although not very thorough in several locations (clearer information on replication would be useful). The presentation is quite nice for some of the figures, although difficult to follow at places and by improving the annotation and clarity of figures the paper would be significantly easier to follow. I hope the further critique points below can help the authors develop the manuscript.

The reviewer raises a valid point. Ideally, these types of analysis should be preferably done by cultivating Asgards and then studying and comparing their features to those of their eukaryotic counterparts. However, laboratory cultivation of these Asgards is so far extremely challenging, despite many attempts by many research groups. There is a single documented case where an Asgard was cultivated in the lab and this took more than a decade (11-12 years). Therefore, the currently only possible way to study these organisms at the moment is to study the functions of their expressed proteins in vitro and compare these with those of eukaryotes. In addition, eukaryotic and asgard actin are highly homologous in sequence and in function, and we strongly believe that conclusions drawn from our experiments are valid and relevant.

The NMR structure appears poor and needs refinement (many Ramachandran outliers) and/or a proper explanation of the data. Some of the structural features described are quite subtle and the structure may not be accurate enough to have confidence in them.

We have refined our structure using CNS and there are no longer any Ramachandran outliers. The structural features are still the same and we are confident with our structural data is robust. We have also submitted a pdb together with this resubmission.

There are 3D structural and sequence-length differences noted between regions of lokiarchaeota- and Heim-profilins, but the data shown does not demonstrate whether these are conserved and characteristic differences between the candidate phyla profilins, as implied, or whether there is significant variability within each taxon. For example, showing well-analyzed alignments, from a large, diverse and balanced sequence set, and corresponding molecular phylogeny of the profilins from across Asgard organisms and eukaryotes would enable the conservation of these features to be identified - these are mentioned later in the manuscript, but data are not shown, apart from an unclearly annotated alignment Supp fig 11..

We agree with the reviewer that the evolutionary analysis of the profilins and of the N-terminal extension was not thoroughly established. In this revision, we performed a robust phylogeny of the profilins and investigated the phylogenetic distribution of the N-terminal extension, which is shown in a new supplementary figure. This analysis revealed that 12 (out of 256, 4.6%) profilin homologs carried an N-terminal extension longer than 5 amino-acid residues. The alignment of the N-terminal extensions is shown in another supplementary figure.

Even if sequence is poorly conserved, the groups should still be resolvable and, particularly, the conserved differences in length of the various regions should be detectable. Are there particular conserved residues that appear, which contribute to the structural differences noted? Are these features predicted to be conserved in the ribosomal trees of the phyla? What is the conservation of length and identity of residues in the N-terminal extension? These would substantiate the more general claims being made about conservation of differences amongst the profilin groups in the major Asgard taxa.

The profilin sequence itself is well conserved and its analysis yields a robust tree, and at the same time reveals a complex evolutionary history, whose fine analysis is beyond the scope of this paper. However, the tree clearly reveals that the N-terminal extensions though not conserved for all members are spread out throughout the tree, suggesting that the gain of these N-terminal extensions could be the product of independent events. We adapted the text accordingly. The results are reported in the Results section and summarized in the Discussion part

Ln 97. And 101. The data in Fig2C/G appear indistinguishable from normal actin (2a/f) visually, but these sentences suggest a slight difference. Clarify description.

We have corrected this sentence and made a new figure 2 to clarify this point

Ln 100. Fig 2f/g The activity on rabbit actin appears very low compared to HeimActin, as might be expected. This should be admitted/noted. How does it generally compare to Loki or other heterologous profilins?

We have added a sentence to note this and further clarify the difference/similarities

Ln 103. Why was the AB_125 actin (fragment) used? As it is not a complete sequence, if its remaining sequence differs from LC3, it cannot be compared properly as a deletion of the C-term (Ln 112). Also, what would be the purpose of deleting the C-terminal 35 amino acids? The way AB_125 protein data is interpreted/discussed should be significantly revised.

We found serendipitously that the AB_125 (Archaea) expressed Actin that was shorter than that from heimdall LC3. This sequence was complete as annotated in the data base and it was not shortened by us. We barely name it C-terminal deletion as it did not contain the last 35 amino acids. The rest of the sequence has a 100% identity. We have provided the sequence for both AB-125 and heimdall LC3 actin in a new S5.

Ln 106. Were the SDS-PAGE conditions reducing or non-reducing? Clarify these conditions in the relevant figure legends and methods, and why they were chosen. In sufficient data are included to determine whether the protein is running as a monomer, dimer or other multimer under the various conditions.

The SDS-PAGE contained reducing agents standard practice is always to run SDS-PAGE with a reducing agent (in that case DTT), which is the reason why we didn't think it was necessary to mention this. The information is however available in the material and methods section. **The samples were also boiled for 10 mins in SDS loading buffer containing DTT before the SDS-PAGE was done. Our size exclusion in the presence of reducing agents indicates that the protein runs as monomer (new S5). We stated in the material and method section the conditions which the SDS and SEC was done.**

Ln 111. Supp Fig 5 data are not well analyzed and the molecular weight claim for heimActin (44 kDa) is weak. Interpolation from a fit to a standard curve for peak maxima of marker proteins is needed as a basic analysis of the supp fig 5 data.

We admit that determining the absolute molecular from such an analysis is weak. However, we have repeated the SEC in the presence of 100mM DTT and the heimActin peak did not differ from that done without. This together with the MALDI-TOF data (new S5) confirms the identity of the heimActin.

Here, also provide the reader with the calculated MW from the sequence, as a comparison. But MW cannot be strictly confirmed by SEC, which is also subject to shape effects and calibration requirements like SDS-PAGE. It is important to show reducing versus non-reducing SDS-PAGE or SEC for proper comparisons to be made. MW can be determined by SEC-MALS or ES-MS, or other more direct methods.

We have also added the sequence of the heimActin and short variant in the new S5 together with the molecular weight of heimActin (indicated in the figure legend). We have provided an additional SEC in the presence and absence of reducing agents in a new S5. In addition, we performed MALDI-TOF mass spectrometry from purified heimActin all confirming the identity of heimActin.

pp4-5, and pp6-8 Split-up the two enormous paragraphs, that make the points hard to follow.

We have rewritten parts here and we believed it is much easier now.

Ln 117. The profilins do not necessarily appear 'efficient' based on the presented data. Quite a moderate effect is seen in SuppFig4a-b, j-m. (also, please note the actin

concentration somewhere here for reference). Given this moderate effect, the experimental variance (errors) should ideally be included in quantified data based on identical replicate polymerization experiments (in panels j-m). Here I also found the bar graphs inappropriate for the type of data - some sort of scatter or line plot would be much clearer, with y-axis as percent of total protein. If showing only the gels or single quantification for a non-quantified basic test, it would be appropriate to include higher concentrations of profilin, to show (presumably) a more complete retention of actin in the supernatant.

This is a comparative analysis of the profilins and each experiment was done back to back. The differences we see are clear within this back to back comparison. These co-sedimentations were done to compliment another analysis done in this work. We have done multiple analysis (pyrene, NMR, TIRFM and ITC) to show the differences between the profilins. The concentrations of actin were mentioned in the materials and method section and have now been added to the figure legend. The experiments in this panel were done multiple times. We have now reported errors in our new plot. We have also plotted our data in a line plot.

Ln 126 Sup Fig 4d-e The results indicate that there is no detectable influence of these profilins on rabbit actin polymerization (pelleting). The claim that these profilins interact with monomeric actin appears very weak and would require a stronger case.

Again, these results are complementing the other results seen in the pyrene assay, TIRF microscopy which indicates that profilins influence rabbit actin polymerization. We have provided additional TIRFM (video clips) showing clearly the difference in speed of rabbit actin monomer addition to the growing filaments in the presence the different profilins.

Ln 135 Supp Fig 6/7. HSQC changes look minimal upon titration of actin and do not appear consistent with specific binding to HeimActin. Is there titration data with different concentrations of HeimActin to show the chemical shift transitions in the putative intermediate binding states?

We did note that the effect on heimProfilin appears weaker compared to the deleted mutant. Also, we could not perform these assays at higher concentrations of actin as it will aggregate and thus cannot be concentrated to concentrations that will give a good saturation curve. However, we have made amino acid point mutagenesis from four of this positions and determined their effect on rabbit actin polymerization in TIRFM. We did see significant differences

in the polymerization speed indicating that these subtle changes seen between heimaActin and heimprofilin/deleted mutant profilin indeed influence the rate of rabbit actin polymerization (new Fig 3).

Ln 145. These lists of interacting residues are not helpful in comparing profilin to eukaryotic actin. It would need a figure, and the residues hint they could be very different. This observation should not be minimized or hidden – Supp Fig 6 does not compare these two as this section suggests (Ln 148) – revise to avoid the mistaken implication.

As we stated above, we selected four position from this list and mutated the amino acids into alanine. We then tested their effect on eukaryotic actin polymerization. We observed changes in the speeds of eukaryotic actin polymerization as well as the length of polymerized actin (new Fig 3).

Ln 147. Were >> Where ***done!***

Ln 151 Are polyproline regions found in Heimdallarchaeota?

We could not find proteins similar to VASP from which the present polyproline peptide was derived in heimdallarchaeota. However, annotation of the Asgard are still an on-going project from different labs and at the moment we do not know.

Ln 157. (fig 3) These interactions are weak and, more concerning, they don't appear to have a defined stoichiometry (the binding model is not clear). The native protein has a significantly weaker affinity than the truncated, and the chemical shift perturbations are also weak in Supp Fig 9 and show no particular concentration of shifts to one region – furthermore, they do not show why the native and truncated proteins would differ. These findings suggest binding is non-specific, or non-physiologically relevant, so relevance of these are in doubt.

Yes, it is true that the heimProfilin has a weaker binding affinity for the polyproline than the deletion mutant and this is what we trying to show. The affinity for the deletion mutant from our ITC is 0.3mM and is well within what has been estimated for eukaryotic profilin/polyproline binding (0.09-0.25mM). For weaker interaction of this sort only the Kd can be reliable obtained from ITC (the binding becomes a typical saturation curve). It is not true that the interactions are unspecific. If this was the case, you will see chemical shift changes randomly

and likely moving in similar direction. This is not what we see. In fact, for residues undergoing chemical shift changes, their directionality follows what we have previously reported and typical for protein ligand interaction (doi.org/10.1021/acs.biochem.7b00965). The chemical shift graph plotted on the structure shows that these shifts are more concentrated in the 3D than in the sequence. We do not expect that concentration of these shifts at the sequence level rather in the 3D. The region experiencing the shift change is the binding site of polyproline and is similar to that for human Profilin/polyproline binding.

Ln 193 is not enough information or data provided on ThorProfilin to determine any features of binding to polypro or its significance. (Fig 5). The role of the ThorProfilin N-terminal extension, where a difference was observed for Heimprofilin, remains unclear and appears inconsistent with the Heim

We have now performed a concentration dependent change for the interaction of thorProfilin and deletion mutant thorProfilin with polyproline (new Fig.6). From our analysis, we see that the deletion mutant has a higher affinity (0.2mM) as compared to the wild type thorProfilin (0.5mM). The significance of this section was to show that other profilins contain extensions that might influence their polyproline binding. And this is what we are reporting

Ln 194 Thro >> Thor **done**

Ln 207. Does the lack of binding to the apparent full-length protein with polypro and PIP indicate this region is not expressed in the archaeon, or that these putative interactions are not physiologically relevant? This section was confusing - what interaction is tighter, given previous sentence said there was no binding (Ln 211)?

We think that the lack of or weaker interaction to the full-length protein could be a regulatory mechanism where a third-party protein might be involved. These are all speculation but we wrote previously thus "We propose a model where modification of the extended N-terminal or interaction with third-party proteins and molecules, causes heimProfilin to behave similarly to Δ N-heimProfilin. This will allow for PIP₂ and polyproline interactions. Concurring or subsequent remodeling would flip heimProfilin to an actin modulating state, allowing for actin polymerization regulation. Modification or interaction with third-party proteins would then be able to reset profilin to the first step of the cycle'

The tighter here is based on the premise from NMR point of view where tighter binders could result in an atom experience fast chemical shift which does not necessarily result in change in position as we observed for a few residues. To

investigate this type of situation we employed the strategy described in the text to determine the relaxation rates. We have added "also" in the text to clarify the sentence.

Ln 227. The clustering of shifted residues indicates 'a potential binding interface for PIP2'. Would such clustering imply that there is no such interface for the previous PolyPro interaction that show limited or no clustering? Are the clusters this located in a homologous region to PIP binding to other profilins?

The polyproline also show clustering and is what we display in the 3D structure (S9). We did not discuss this as such because this site has been mapped previously from other profilins. This cluster site for PIP2 indeed is located in homologues region found in other profilins and we mentioned a few residues in homologues positions between eukaryotic (K69 and K90), loki (K60 and K71) and heimProfilin (K110 and K146).

Reviewers' comments:

Reviewer #1 (Remarks to the Author):

The points I have raised after reviewing the original version of the manuscript have been addressed to my satisfaction. Therefore, and because of the relevance of the work in general, I recommend publication in *Commun Biol* of the current form of the manuscript.

Reviewer #2 (Remarks to the Author):

The paper is improved and has addressed most of the technical concerns of the reviewers, though the writing and structure should be improved, particularly the results section, which have large blocks of text that would need to be properly paragraphed and sectioned (with titles ideally). It is hard to follow any logical sequence of points that the manuscript should make.

However, the significance and validity of several aspects of this study highlighted by the authors remains low in this revision. Since polyproline binding is very weak for the full-length Heim-profilin, there is insufficient evidence to support its claim as a specific physiologically significant interaction. Heim-profilin binding to polyproline is one of the main claims of significance in the abstract (the Heim-profilin being supposedly different to other Asgard profilins in binding to polyproline, which were not tested here). The N-term deletion mutant does bind a little tighter, but, rather than being a regulatory region, this suggests that the apparent N-terminal extension could be a mis-annotation of the start site for the heim-profilin ORF. Consistent with this, the alignments and phylogeny that the authors have now included show that the N-term extension is very poorly conserved and scattered amongst Asgards, and in only ~4% of have them. Many of the results presented rely on comparison of the 'full-length' and truncated versions of heim-profilin, and the authors assume that the N-terminus must be a regulatory region and not a mis-annotation. This leaves the validity and relevance of the studies of the N-terminal region in serious doubt. Whether the N-term extension is present or not in the theoretical native protein, the low conservation of the N-terminal extension, and the different polyproline binding results obtained for thor-profilin, also show that it cannot be broadly applied to Asgard or Heimdallarchaeota profilins.

The significance of the Heim-actin work is greater than the weak effects on rabbit actin, but this is presented almost as a side observation. Are polyproline motifs and PIPs expected to be present in Heimdallarchaeota?

The abstract contains an apparent contradiction; Line 21 ('Nterm region unseen in other Asgards', and then (Line 23); "Asgards seemed to possess profilin with similar extended region"?

Line 106. Can the authors indicate the sequence identity between the LC3 and AB_125 actin sequences?

Reviewers' comments:

Reviewer #1 (Remarks to the Author):

The points I have raised after reviewing the original version of the manuscript have been addressed to my satisfaction. Therefore, and because of the relevance of the work in general, I recommend publication in Commun Biol of the current form of the manuscript.

Reviewer #2 (Remarks to the Author):

The paper is improved and has addressed most of the technical concerns of the reviewers, though the writing and structure should be improved, particularly the results section, which have large blocks of text that would need to be properly paragraphed and sectioned (with titles ideally). It is hard to follow any logical sequence of points that the manuscript should make.

We agree with the reviewer that the readability could be improved. We have restructured the text with subheadings that describe the main conclusions of the respective sections.

However, the significance and validity of several aspects of this study highlighted by the authors remains low in this revision. Since polyproline binding is very weak for the full-length Heim-profilin, there is insufficient evidence to support its claim as a specific physiologically significant interaction. Heim-profilin binding to polyproline is one of the main claims of significance in the abstract (the Heim-profilin being supposedly different to other Asgard profilins in binding to polyproline, which were not tested here). The N-term deletion mutant does bind a little tighter, but, rather than being a regulatory region, this suggests that the apparent N-terminal extension could be a mis-annotation of the start site for the heim-profilin ORF. Consistent with this, the alignments and phylogeny that the authors have now included show that the N-term extension is very poorly conserved and scattered amongst Asgards, and in only ~4% of have them. Many of the results presented rely on comparison of the 'full-length' and truncated versions of heim-profilin, and the authors assume that the N-terminus must be a regulatory region and not a mis-annotation. This leaves the validity and relevance of the studies of the N-terminal region in serious doubt. Whether the N-term extension is present or not in the theoretical native protein, the low conservation of the N-terminal extension, and the different polyproline binding results obtained for thor-profilin, also show that it cannot be broadly applied to Asgard or Heimdallarchaeota profilins.

We followed the reviewer's advice and reworked the structure of the discussion, putting forward first the results that do not depend on the N-terminal extension, then discussing the role extensions, weighing the likelihood of them being artifacts, and discussing the implications of both scenarios. We have also toned down the regulatory role of the extensions proposed here. However, we believe that the data supports a physiologically significant interaction for both the full-length and the N-term deletion mutant.

The significance of the Heim-actin work is greater than the weak effects on rabbit actin, but this is presented almost as a side observation. Are polyproline motifs and PIPs expected to be present in Heimdallarchaeota?

This work was more focused on the comparison of Asgard and eukaryotic profilins. However, we found some interesting aspects of heimActin but decided not to stress too much on it as we do not yet have enough information on its structure (3D structure). Specific polyproline proteins such as ENA/VASP have not been identified in Asgard archaea but Archaea are known to encode IF-5A which in eukaryotes alleviates ribosomal stalling at polyproline motifs, and contain many polyproline motives. Archaea do contain PIP2 with similar head groups as those in eukaryotes but different tails. Following the reviewer's suggestion, we have highlighted the results from the experiments with heimActin in the discussion.

The abstract contains an apparent contradiction; Line 21 ('Nterm region unseen in other Asgards', and then (Line 23); "Asgards seemed to possess profilin with similar extended region"?

We have corrected this statement

Line 106. Can the authors indicate the sequence identity between the LC3 and AB_125 actin sequences?

This has been indicated

REVIEWERS' COMMENTS:

Reviewer #4 (Remarks to the Author):

The latest version of the paper presents a more balanced view of the results, and the authors have improved the readability significantly by setting out the results in parts. I'd still suggest the paper is well edited as there were locations that could be clearer, or have typo. e.g. spelling of motifs, rather than motives.

e.g. The abstract has some redundancy in it. The N-terminal extension is introduced in the second last sentence in a confusing way. I'd delete this sentence, and leave this to the discussion, or alternatively briefly describe the +/- N-term results so the context of this statement regarding the extension is clearer.

It may be worth instead summarizing in the abstract, and in the paper, the likelihood that most Asgard profilins (including the previously studied one that was claimed not to bind significantly) may interact with polypro motifs (weakly), and whether this is likely to be physiologically relevant in Asgards i.e. do the poly-pro motifs the authors show in the latest version of the paper exist within proteins that are related to the relevant mammalian poly-pro motifs? And: importantly by comparing the affinity of binding this to the affinity previously observed for mammalian profilin-polypro interaction.

i am satisfied that technical considerations have now been addressed by the authors, and if the editor is satisfied that the authors have addressed these final comments, or any other revisions to the manuscript, I would recommend publication without further external reviews

Reviewer #5 (Remarks to the Author):

The manuscript brings more information about the early evolution of eukaryotes by bringing evidence of a eukaryotic protein family shared by eukaryotes and Archaea. The authors already addressed the points raised by other reviewers of the first round of revision. The study leaves open questions about the subject, but I think this is normal in science. I particularly think that the manuscript is not focused on evolution, even though its relevance lies inside the field. This could be improved.

Reviewer #4 (Remarks to the Author):

The latest version of the paper presents a more balanced view of the results, and the authors have improved the readability significantly by setting out the results in parts. I'd still suggest the paper is well edited as there were locations that could be clearer, or have typo. e.g. spelling of motifs, rather than motives.

Done

e.g. The abstract has some redundancy in it. The N-terminal extension is introduced in the second last sentence in a confusing way. I'd delete this sentence, and leave this to the discussion, or alternatively briefly describe the +/- N-term results so the context of this statement regarding the extension is clearer.

We are thankful to the reviewer for pointing this out. We have deleted the second last sentence since this is discussed in detailed in the discussion part of the manuscript.

It may be worth instead summarizing in the abstract, and in the paper, the likelihood that most Asgard profilins (including the previously studied one that was claimed not to bind significantly) may interact with polypro motifs (weakly), and whether this is likely to be physiologically relevant in Asgards i.e. do the polypro motifs the authors show in the latest version of the paper exist within proteins that are related to the relevant mammalian poly-pro motifs? And: importantly by comparing the affinity of binding this to the affinity previously observed for mammalian profilin-polypro interaction.

We have added a few sentences summarizing this in the paper (colored coded in yellow). As this is an open discussion and also due lack of space in the abstract, we did not discuss it in the abstract. Yes, the polyproline motifs we reported are also related to mammalian polyproline motifs. However, the affinities of only a few have been determined in mammals. We previously highlight this in the main text and have reiterated it again the discussion.

i am satisfied that technical considerations have now been addressed by the authors, and if the editor is satisfied that the authors have addressed these final comments, or any other revisions to the manuscript, I would recommend publication without further external reviews

Reviewer #5 (Remarks to the Author):

The manuscript brings more information about the early evolution of eukaryotes by bringing evidence of a eukaryotic protein family shared by eukaryotes and Archaea. The authors already addressed the points raised by other reviewers of the first round of revision. The study leaves open questions about the subject, but I think this is normal in science. I particularly think that the manuscript is not focused on evolution, even though its relevance lies inside the field. This could be improved.

We appreciate this comment from the reviewer. It is true that this manuscript is focused more on elucidating the structural and biochemical aspects of Asgard archaea profilins/actin and its comparison to their eukaryotic counterpart. This is mainly because the evolutionary aspect has been heavily dealt with in the original metagenomics papers.